# TRAINING ON THE TEST TASK CONFOUNDS EVALUATION AND EMERGENCE

**Ricardo Dominguez-Olmedo** [,1]**, Florian E. Dorner**[1,2]**, and Moritz Hardt**[1]

[1]Max Planck Institute for Intelligent Systems, Tübingen, and Tübingen AI Center [2]ETH Zürich

## ABSTRACT

We study a fundamental problem in the evaluation of large language models that we call *training on the test task*. Unlike wrongful practices like training on the test data, leakage, or data contamination, training on the test task is not a malpractice. Rather, the term describes a growing set of practices that utilize knowledge about evaluation tasks at training time. We demonstrate that training on the test task confounds both relative model evaluations and claims about emergent capabilities. We argue that the seeming superiority of one model family over another may be explained by a different degree of training on the test task. To this end, we propose an effective method to adjust for the effect of training on the test task on benchmark evaluations. Put simply, to fine-tune each model under comparison on the same task-relevant data before evaluation. We then show that instances of emergent behavior disappear gradually as models train on the test task. Our work promotes a new perspective on the evaluation of large language models, with broad implications for benchmarking and the study of emergent capabilities.

## 1 INTRODUCTION

The machine learning community has long recognized certain clear violations of the benchmarking protocol. Training on the test set is the most notorious among them (Duda & Hart, 1973; Hastie et al., 2017). Data leakage (Kapoor & Narayanan, 2022) and data contamination (Roberts et al., 2023; Jiang et al., 2024) are closely related problems linked to the rise of massive web-crawled training datasets. Researchers can all agree that test data should never appear in the training set.

But it's been much less clear what to do about legitimate attempts to bring training closer to evaluation. There is an obvious a gap between next token prediction at training time and tasks, such as reasoning and question answering, at test time. Ongoing research and engineering efforts, in fact, aim to narrow precisely this gap (MetaAI, 2024). Why shouldn't training be informed by knowledge about the downstream test tasks? What's an unfair advantage for some may be the feature of others.

In this work, we group strategies to utilize task knowledge at training time under the umbrella term of *training on the test task*. Examples of training on the test task include the use of instruction-tuning data or question answering templates during pre-training (Bai et al., 2023; StabilityAI, 2023; Groeneveld et al., 2024). Models may also implicitly train on the test task when their pretraining data is selected through ablations on downstream benchmark evaluations (Gemma et al., 2024; MetaAI, 2024). We work from the premise that training on the test task is acceptable–or, at least, unavoidable.

In a nutshell, we show that training on the test task strongly confounds model comparisons across different scales and model families. Perhaps counterintuitively, we propose to mitigate the effects of training on the test task on benchmark evaluations by doing *more* of it. We show that we can effectively level the playing field by giving each model the same, sufficient task-specific fine-tuning before evaluation. This adjustment restores cleaner log-linear scaling and makes capabilities predictable based on much smaller model scales.

### 1.1 OUR CONTRIBUTIONS

We introduce the term training on the test task to group a growing repertoire of practices that utilize knowledge about evaluation tasks at training time. We study its impact on present-day benchmark

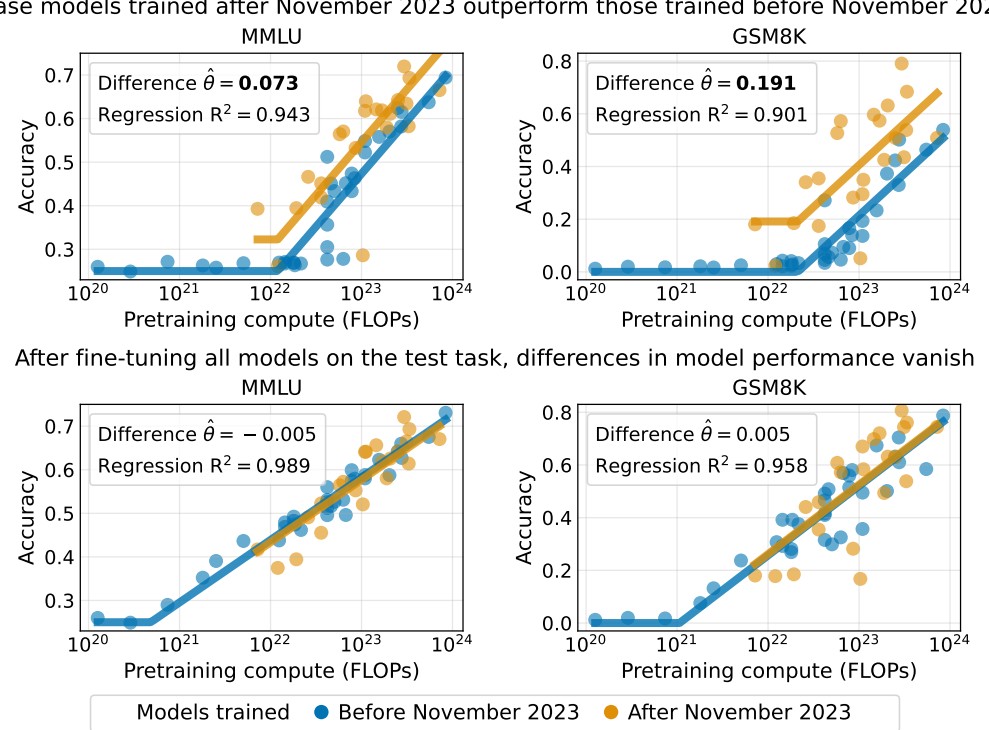

Figure 1: MMLU and GSM8K scores of 56 base models, with model sizes ranging from 70M to 70B parameters. Solid lines correspond to the regression fit of $A = \alpha \max(0, \log C - c_e) + \theta N + r$, where $A$ is accuracy, $C$ is pretraining compute, $N$ is whether the model was trained after November 2023, and $r$ is random chance accuracy. The coefficient $\theta$ denotes the average improvement of models trained after November 2023 when controlling for pretraining compute. Bold indicates statistical significance with $p$-value $< 0.05$. *(Top)* We hypothesize that training on the test task confounds benchmark evaluations, resulting in newer base models substantially outperforming older ones. *(Bottom)* We propose to adjust for differences in test task training by fine-tuning all models on the same, sufficient amount of task-specific data before evaluation. After fine-tuning on the test task, differences in benchmark performance between older and newer models vanish.

evaluations by critically examining the performance improvements of recent language models. Our analysis spans 56 different language models and two major active benchmarks, MMLU and GSM8K.

We start in Section 2 by dividing models into those trained before November 2023 and those trained after. We find that for the same amount of pretraining compute, newer models strongly outperform older ones, on average by 7 percentage points in MMLU and 19 points in GSM8K. We then fine-tune all models on the same amount of task-specific data before evaluation. After fine-tuning on the same task data, newer models no longer outperform older ones. Rather, their performance equalizes. See Figure 1. This outcome suggests that the main difference between newer and older models is the extent to which they train on the test task.

We propose a simple and effective method to adjust for the effect of training on the test task on benchmark evaluations. Put simply, to fine-tune each model on the same, sufficient amount of task-specific data before evaluation. To validate our method, we demonstrate its effectiveness in a controlled setting: we take the older models and fine-tune them on the test task. Remarkably, this recreates the kind of performance differences observed between newer and older models, further suggesting that training on the test task explains the improvements of newer models. We then show that we can undo the advantage of the fine-tuned models over the other models by further fine-tuning all models on the test task (Section 3.1, Figure 3).

Next, we provide evidence that training on the test task may be a more dominant factor in benchmark performance than data contamination. To argue this point, we consider ARC and HellaSwag. Here, at first, there appears to be no sign of newer models having an unfair advantage over older models. But after reformulating these benchmarks as MMLU-style multiple choice question answering tasks (MCQA), we see the same confounded results as for MMLU (Section 3.2, Figure 4). This suggests that the improvements of newer models on MMLU are likely not because of memorization of specific testing data, but rather due to an improved ability for MCQA tasks.

Then, we show how training on the test task distorts model family comparisons. Certain model families appear markedly superior to others before adjusting for test task training, but not after adjustment (Section 4.1, Figure 6). We then demonstrate how training on the test task has inflated the perceived progress made by recent model families. After adjusting for its effect, newer models only modestly improve the Pareto frontier of model performance relative to pre-training compute.

Finally, we demonstrate that training on the test task has profound implications for the study of emergent capabilities. The phenomenon of emergence disappears gradually as the amount of training on the test task grows (Section 5). Specifically, we can make capabilities visible and predictable from much smaller model scales, recovering cleaner log linear-scaling. Importantly, our adjustment also works in cases, like MMLU, where previous purported explanations of emergence, such as the choice of evaluation metric, do not suffice.

Our work calls for a major reorientation of large language model evaluation. Model comparisons and claims of emergence are strongly confounded by the choice of training data relative to the test tasks. When comparing models with different pre-training data, our recommendation is to give each model the same sufficient amount of fine-tuning on task-relevant data before evaluation.

## 2    ADJUSTING FOR TRAINING ON THE TEST TASK

We choose MMLU (Hendrycks et al., 2020) and GSM8K (Cobbe et al., 2021) as a case study for investigating training on the test task in active benchmarks. MMLU tests for world knowledge, whereas GSM8K tests multistep mathematical reasoning. These two benchmarks are arguably the most influential of the 2022-2024 period under study. They are also included in the Hugging-Face (HF) Open LLM Leaderboard v1 (Beeching et al., 2023), a popular leaderboard that evaluates and ranks models with publicly available weights. We evaluate models using LM Evaluation Harness (EleutherAI, 2024), in identical fashion to the HF leaderboard[1].

We evaluate 56 base models, ranging in size from 70M to 70B parameters. See Appendix B.1 for the full list. The HF leaderboard's FAQ makes the distinction between "base pretrained models" and instruction-tuned or chat models, arguing that this is necessary to ensure fair model comparisons. We select models that are categorized as "pretrained". We check that the technical report of each of the selected models makes no mention of the model being fine-tuned. We only consider models for which the number of training tokens is known. This allows us to estimate the total amount of pretraining compute in FLOPs as $C \approx 6 \cdot N \cdot D$, where $C$ is pretraining compute, $N$ is the number of model parameters, and $D$ is the number of training tokens.

While we focus primarily on MMLU and GSM8K due to their prominence, we find that the issue of training on the test task extends beyond these two benchmarks. Specifically, in Appendix E, we evaluate and discuss the impact of training on the test task for five additional benchmarks: MMLU Pro (Wang et al., 2024), GPQA (Rein et al., 2023), BBH (Suzgun et al., 2023), MuSR (Sprague et al., 2023), and MATH Level 5 (Hendrycks et al., 2021), which form the OpenLLM Leaderboard v2 (Fourrier et al., 2024a). Furthermore, in Appendix G we conduct similar experiments for an additional 36 instruction and chat models. We observe that our findings generalize remarkably well to instruction and chat models.

**Recent models outperform older ones given the same pretraining compute.**    We evaluate models on MMLU and GSM8K, and plot benchmark accuracy against pretraining compute in Figure 1 top. We observe that performance correlates with pretraining compute for both benchmarks. However, on the surface it appears that models released after November 2023 better leverage pretraining compute. For a given compute budget, newer models tend to attain better benchmark performance.

---

[1]We also evaluate MMLU and GSM8K 5-shot, ARC 25-shot, and HellaSwag 10-shot.

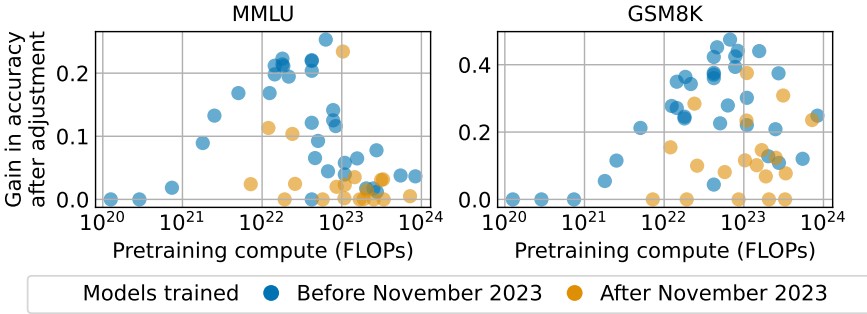

Figure 2: Models trained before November 2023 tend to benefit more from fine-tuning on task data.

We selected November 2023 as the temporal cutoff for our analysis because the technical reports of models released from late 2023 onward start referencing certain pre-training practices that may amount to training on test task. For example, Qwen (Bai et al., 2023), Olmo 1.7 (Groeneveld et al., 2024) and MAP Neo (Zhang et al., 2024) explicitly include instruction data during pretraining. StableLM 2 (StabilityAI, 2023) reformulates some of its pretraining datasets to better resemble downstream tasks such as question-answering. More subtly, the pretraining data mixtures of Gemma (Gemma et al., 2024) and Llama 3 (MetaAI, 2024) were determined through extensive ablations on downstream benchmark evaluations. We validate that our findings are robust to adjusting the temporal cutoff by a few months; see Appendix D.1 for details. Choosing specifically the month of November as the cutoff is therefore not critical for our analysis.

This raises an important question: Do newer models outperform older ones mainly because newer models trained more on the test task? At first sight, an answer seems elusive. After all, the pre-training data of most recent models is not publicly available. Retraining all models with the same training data and compute budget would be both infeasible and cost prohibitive. In the next section, we propose a way to get at the answer by adjusting for the effect of training on the test task.

## 2.1 ADJUSTING FOR TRAINING ON THE TEST TASK BY TRAINING ON THE TEST TASK

We propose to adjust for differences in test task training by fine-tuning all models on the same, sufficient amount of task-specific data before evaluation. To do so, we need a source of task-specific data for each of the tasks we consider. For multiple choice questioning answering, we use the auxiliary training set accompanying the HF MMLU repository[2]. This training set is not an i.i.d. split of MMLU. Instead, it consists of the training sets from other multiple-choice question-answering benchmarks, comprising approximately 100,000 training examples and 30 million tokens. For mathematical reasoning, we combine MetaMathQA (Yu et al., 2023b) and Orca-Math (Mitra et al., 2024), totalling approximately 600,000 training examples and 200M tokens. We fine-tune models for three epochs using standard hyperparameter choices, see Appendix B.2. The amount of compute required for fine-tuning is minimal compared to models' pretraining compute.

We plot model scores on MMLU and GSM8K after fine-tuning in Figure 1 (bottom). We observe that after fine-tuning on task relevant data, both newer and older models follow remarkably similar scaling trends. That is, newer models no longer appear to outperform older models.

Remarkably, we observe that older models tend to benefit much more from fine-tuning on task-relevant data compared to newer models, see Figure 2. The improvements in older models are striking, often leaping from random chance accuracy to double-digit gains in accuracy. In contrast, fine-tuning provides comparatively little benefit to newer models. This observation suggests that newer models have already been exposed to a substantial amount of task-relevant data, making additional fine-tuning less impactful.

A potential concern is that our observations might result from our fine-tuning hyperparameters being systematically more favorable to older models. We verify that this is not the case by conducting a robustness check on the fine-tuning hyperparameters, see Appendix B.3.

---

[2]https://huggingface.co/datasets/cais/mmlu

## 2.2 QUANTIFYING PERFORMANCE DIFFERENCES BETWEEN NEWER AND OLDER MODELS

We draw inspiration from scaling laws (Kaplan et al., 2020) in how we model benchmark accuracy $A$ to scale log-linearly with pretraining compute $C$. To account for emergence (Wei et al., 2022b), we assume that models perform at the task's random chance accuracy $r$ up to scaling to some point of emergence $c_e$. We let the variable $N$ denote whether a model was trained after November 2023, and regress the model

$$A = \alpha \max(0, \log C - c_e) + \theta N + r + \epsilon, \tag{1}$$

where $\alpha$, $\theta$ and $c_e$ are the fit's parameters, and $\epsilon$ is random noise. We focus on the coefficient $\theta$, which corresponds to the average difference in benchmark performance between newer and older models after controlling for pretraining compute. We fit the model in Equation 1, and report the regression coefficient $\theta$ in Figure 1. We obtain $R^2 > 0.9$ for all model fits. We use clustered standard errors to compute statistical significance, treating each model family as a separate group.

Before adjusting for test task training, the estimated difference in performance $\hat{\theta}$ between newer and older models are statistically significant, positive, and large. Specifically, recent models outperform older ones on average by over 7 accuracy points in MMLU and 19 accuracy points in GSM8K. These are remarkably large differences in benchmark performance. However, after the adjustment, the estimated coefficient $\hat{\theta}$ is both small and not statistically significant. See Figure 1 bottom. That is, conditioned on all models training on the same amount of task-specific data, we find no evidence for a significant difference in benchmark performance between newer and older models.

Therefore, the performance of newer and older models equalizes when all models are exposed to the same amount of task-relevant data. This suggests that the impressive benchmark improvements of newer models are primarily attributable to newer models training more on the test task. We present a causal interpretation of results in Appendix C, outlying the assumptions necessary to establish a causal link between training on the test task and the benchmark improvements of newer models.

## 3 RECREATING DIFFERENCES IN BENCHMARK PERFORMANCE

We have so far established that newer models strongly outperform older models for the same amount of pre-training compute. We now demonstrate how to recreate such differences in performance by actively manipulating how much models train on the test task. We do so in two ways. First, we fine-tune older models on task relevant data (Section 3.1). Second, we reformulate certain test tasks to use MMLU-style multiple choice prompts instead of "cloze" evaluations (Section 3.2). Both experiments recreate the kind of performance differences observed between newer and older models.

These results provide further evidence that the differences in performance between older and newer models are linked to test task training. They also demonstrate how test task training distorts benchmark evaluations. Fortunately, in both cases, we show that fine-tuning models on task-relevant data before evaluation is an effective mechanism for mitigating the bias introduced by training on the test task. In doing so, we systematically validate the proposed adjustment method.

### 3.1 FINE-TUNING ON THE TEST TASK

For this section, we only consider models trained before November 2023. We split the models into two cohorts: a control group and a treatment group. We take these older models as a control group. We then create a treatment group by fine-tuning the control group on the datasets of task-relevant data introduced in Section 2. We only fine-tune models with at least $7 \cdot 10^{21}$ FLOPs, the pre-training compute of the smallest newer model. We fine-tune for a single epoch. We plot in Figure 3 top the benchmark performance of the two cohorts.

Qualitatively, the differences in performance between the control and treatment groups resembles the differences observed between newer and older models, contrast Figure 3 with Figure 1. Quantitatively, the estimated performance gain $\hat{\theta}$ from fine-tuning is similar to the difference between newer and older models estimated in Section 2.2. That is, fine-tuning older models on the test task produces both qualitatively and quantitatively similar confounding to that observed between newer and older models. This results further supports our running hypothesis that newer models are largely equivalent to older models that have trained on the test task. They also demonstrate the large effect

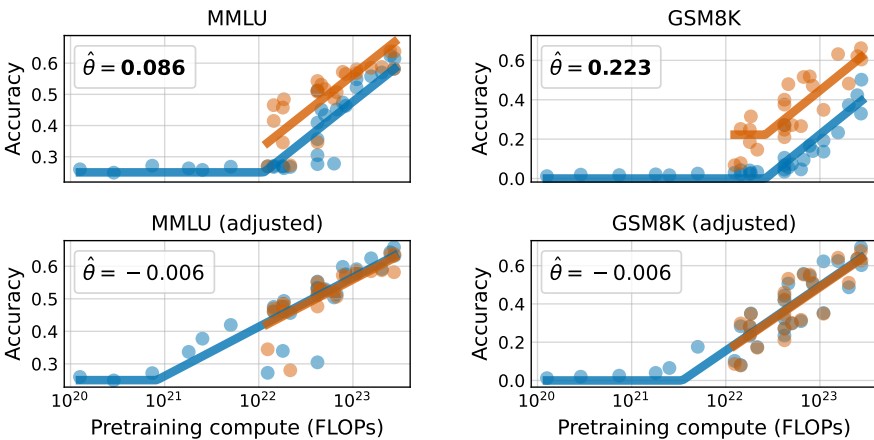

Figure 3: Models trained before November 2023 (●) without fine-tuning and (●) after fine-tuning on the test task. Their difference in benchmark performance $\hat{\theta}$ resembles that between newer and older models. After adjusting by training on the test task, their difference vanishes.

that training on the test task can have on benchmark performance. Note that the gain in performance of the treatment group is slightly larger than the difference in performance between newer and older models. This is to be expected, as all models in the treatment group are fine-tuned on the test task, whereas not all new models may train on the test task.

We then apply our proposed adjustment by further fine-tuning both the control and treatment groups on the test task, see Figure 3 right. After the adjustment, the estimated difference in performance $\hat{\theta}$ between the control and treatment group is both small and not statistically significant. We therefore validate a vital soundness property of the proposed adjustment procedure: after deliberately training some models on the test task, we can undo their advantage over other models by further training all models on the test task.

## 3.2 REFORMULATING THE TEST TASK

In this section, we show that reformulating other benchmarks as multiple-choice question answering tasks leads to similar differences in performance between older and newer models. We consider two additional benchmarks from the HF leaderboard v1: ARC Challenge (Clark et al., 2018) and HellaSwag (Zellers et al., 2019). Similarly to MMLU, ARC comprises grade-school level questions. HellaSwag instead tests for commonsense reasoning. Like MMLU, the questions in ARC and HellaSwag are accompanied by four possible answers, among which the model must differentiate the correct one. ARC and HellaSwag use "cloze" evaluations: a models' answer is taken to be that with the largest completion likelihood given the input question. In contrast, MMLU formulates questions as multiple-choice: all four answer choices are listed, and the model is promoted to pick one.

We first evaluate all models on ARC and HellaSwag using the standard cloze evaluation, and plot their benchmark performance in Figure 4 left. We repeat the statistical analysis of Section 2.2. We find that the estimated difference in performance $\hat{\theta}$ between newer and older models is small and not statistically significant: newer models do not outperform older models on ARC and HellaSwag.

We then reformulate ARC and HellaSwag as MMLU-style multiple-choice questions, and plot the resulting benchmark performance in Figure 4 center. We observe large differences in performance between newer and older models. Specifically, we find the difference in performance $\hat{\theta}$ between newer and older models to be significant, positive, and large, and to be roughly similar in magnitude to that estimated for MMLU in Section 2.2. That is, reformulating the test task as multiple choice question answering leads to qualitatively and quantitatively similar confounding to that observed for MMLU. Therefore, newer models overperform on MMLU likely not because of memorization of specific testing data (i.e., due to data contamination or leakage), but rather due to an improved ability for multiple-choice question answering.

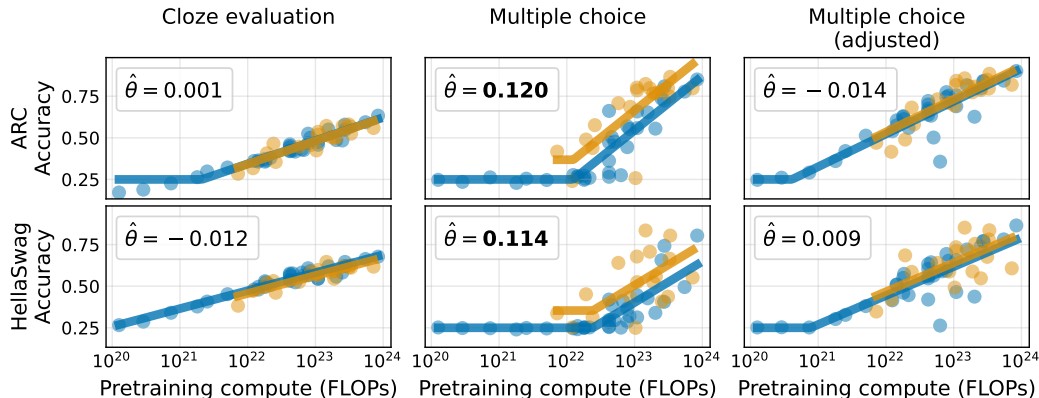

Figure 4: Reformulating ARC and HellaSwag as MMLU-style questions give rise to large differences $\hat{\theta}$ between models trained (●) before November 2023 and (●) after November 2023. After adjusting by fine-tuning on the test task, differences in performance vanish.

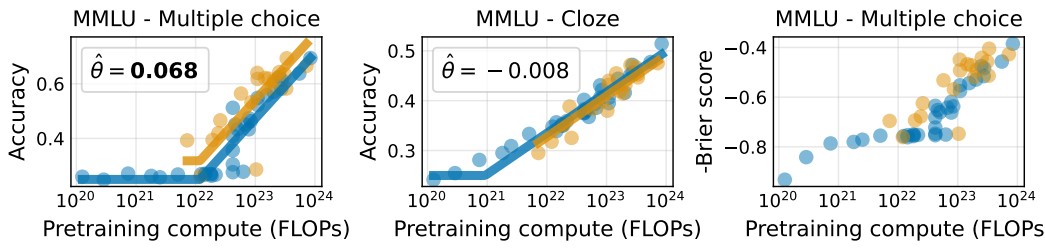

Figure 5: When evaluating MMLU using "cloze" prompts, models trained (●) after November 2023 no longer outperform those trained (●) before November 2023 (*middle*). When using Brier score as the evaluation metric, we still observe sharp improvements in performance (*right*).

Lastly, we adjust for test task training by fine-tuning all models on the MMLU auxiliary training set, and plot their ARC Challenge and HellaSwag scores in Figure 4 right. We no longer find evidence of a large nor a significant difference in performance between newer and older models. Therefore, the proposed adjustment is effective in mitigating the bias introduced by evaluating models via multiple-choice question answering. Notably, performance on ARC and HellaSwag equalizes after fine-tuning on the MMLU auxiliary training set. This indicates that the adjustment data need not closely resemble the test set, but rather the test task.

**What does MMLU test for?** We evaluate MMLU using the "cloze" methodology instead of the usual multiple-choice prompts. We plot the results in Figure 5 center. With cloze evaluations, the difference in performance between newer and older models is both small and not statistically significant. This suggests that the standard MMLU evaluation conflates knowledge-testing with testing a models' ability to answer multiple choice questions[3]. Newer models therefore attain higher MMLU scores than older models largely because they are better at multiple-choice question answering, and not because they necessarily "know more".

## 4 IMPLICATIONS FOR MODEL COMPARISONS

So far, we have shown how training on the test task distorts benchmark evaluations. Next, we examine its impact on the relative comparison of model families (Section 4.1) as well as its implications for accurately measuring progress in model capabilities over time (Section 4.2).

---

[3]For instance, smaller models suffer from particularly strong A-bias for multiple-choice questions (Dominguez-Olmedo et al., 2023).

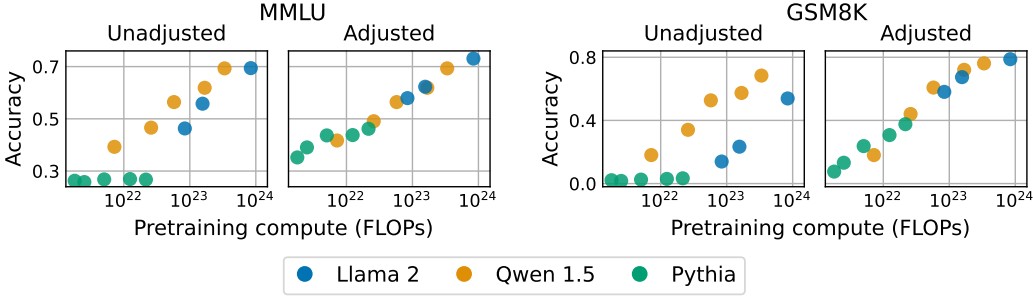

Figure 6: Training on the test task confounds relative comparisons between model families. After adjusting for test task training, none of the three model families appears to be superior.

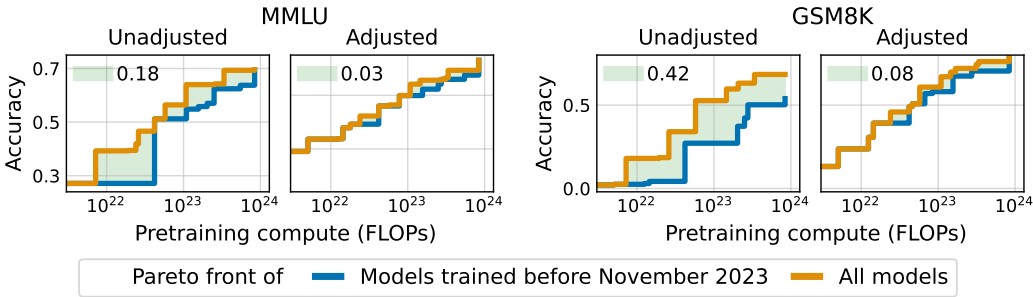

Figure 7: Training on the test task overestimates the improvements made by recent base models. After adjustment, the area of improvement (green) reduces by a sixfold.

## 4.1 COMPARING MODEL FAMILIES

We compare the performance of the Pythia, Llama 2, and Qwen 1.5 model families, which likely train on the test task to very different extents. Pythia was trained on the Pile (Gao et al., 2020), a collection of curated datasets that are unlikely to contain much test task data. Llama 2 was trained mostly on web data, which is reasonable to assume may contain more test task data. Lastly, Qwen 1.5 explicitly includes instruction data in its pretraining mixture, thus likely training on the test task.

We plot the MMLU and GSM8K scores of the three model families in Figure 6, as well as their adjusted scores (i.e., after fine-tuning on task relevant data). Without adjustment, Qwen 1.5 appears to be the superior model family: it Pareto dominates both the Llama 2 and Pythia models. In contrast, all Pythia models perform no better than random chance, making it unclear whether scaling Pythia offers any benefit at all. After adjustment, however, all three model families exhibit remarkably similar scaling trends. Therefore, after correcting for the confounding of test task training, none of the model families appears superior to the others.

Training on the test task therefore profoundly confounds relative model comparisons. Base models are rarely used "as is" and are generally adapted before deployment. Because of the confounding of training on the test task, performance before adaptation may not reliably predict performance after adaptation. It therefore makes little sense to compare base models at face value.

## 4.2 PROGRESS IN MODEL CAPABILITIES

Training on the test task substantially overestimates the progress in benchmark performance per unit of compute achieved by recent model families. In Figure 7 we plot the Pareto frontier of benchmark accuracy against pretraining compute, both for models trained before November 2023 and for all models. We measure progress by considering the area of improvement of the Pareto frontier since November 2023, shaded in green. Without adjustment, the difference between the two Pareto frontiers is large, indicating very substantial progress since November 2023. After adjustment, however, the area of improvement reduces by a sixfold, showing only modest improvements. Therefore,

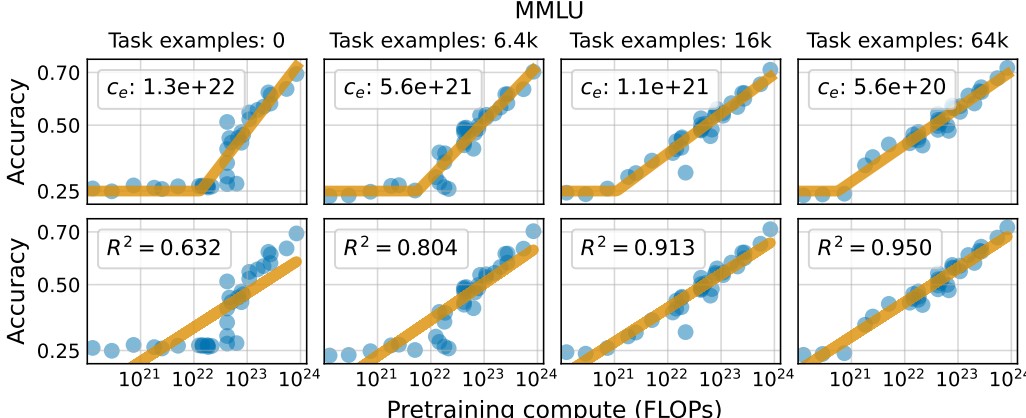

Figure 8: Scaling on MMLU as models increasingly train on the test task. The point of emergence $c_e$ arises at lower scales (*top*). Training on the test task yields cleaner log-linear scaling fits (*bottom*).

training on the test task strongly overestimates the progress in benchmark performance per unit of compute achieved by recent model families.

On the other hand, recent models tend to be trained on more data than Chinchilla compute-optimal (Hoffmann et al., 2022). Given the Chinchilla scaling laws, it is noteworthy that newer, smaller "over-trained" models match the performance of older, larger ones for the same amount of pretraining compute. Since inference and fine-tuning of smaller models is substantially cheaper, recent models can be much more accessible to less well-resourced institutions, with little cost in performance. For example, we find that Llama 3 8B closely matches the performance of Llama 2 70B (both have similar pre-training compute).

## 5 IMPLICATIONS FOR EMERGENCE

Throughout our evaluations, we observe emergent behavior for MMLU and GSM8K: models perform at near random chance up to a certain scale of pretraining compute, followed by relatively sharper improvements in performance at larger scales (Wei et al., 2022b). After training on the test task, however, emergence for MMLU and GSM8K appears to occur at substantially lower scales. We dedicate this section to more closely investigating the relationship between training on the test task and emergence.

**Emergence arises at lower scales with increased test task training.** We consider only models trained before November 2023, as we have established that these models train on the test task less than newer models. We evaluate the models at intermediate checkpoints as we fine-tune them on the datasets of task relevant data introduced in Section 2.1. We fit $\alpha$ and $c_e$ in Equation 1 to the different intermediate checkpoints, and report in Figure 8 top the corresponding points of emergence $c_e$. We find that emergence arises at increasingly lower compute regimes as models train on the test task. For MMLU, models exhibit emergence at around $10^{22}$ FLOPs, the scale of Pythia 6.9B. After training on 64,000 examples, emergence arises around $6 \cdot 10^{20}$ FLOPs, the scale of Pythia 410M. We observe similar results for GSM8K, see Figure 19 in Appendix F.

**Training on the test task yields increasingly better log-linear fits.** The log-linear relationship between pretraining loss and compute is well-established (Kaplan et al., 2020). We observe that training on the test task increasingly recovers log-linear scaling between pretraining compute and benchmark accuracy. Similarly to the earlier section, we evaluate intermediate checkpoints but instead fit log-linear functions in Figure 8 bottom. We observe that the $R^2$ of the fit improves substantially as the models train on more task-relevant data, jumping from $0.63$ to $0.95$ after training on 64,000 examples. Therefore, after training on the test task, almost all the variation in benchmark accuracy is explained by log-linear scaling of pre-training compute. We observe similar results for GSM8K, see Figure 19 in Appendix F.

**Recommendations.** Schaeffer et al. (2024a) argue that emergence appears due to the choice of metric. To mitigate emergence, they suggest considering Brier score instead of accuracy. We observe, however, that emergence for MMLU does not disappear when using the Brier score, see Figure 5 right, nor that of ARC and HellaSwag when framed as multiple-choice questions, see Figure 18 in Appendix F. We discuss two practical solutions to obtain predictive scaling while maintaining accuracy as the evaluation metric.

For MMLU and multiple-choice benchmarks more broadly, cloze evaluations consistently yield smoother and more predictable scaling, even when using accuracy as the evaluation metric. Since the purpose of these benchmarks is knowledge-testing more so than testing multiple-choice answering ability, cloze evaluations are preferable insofar predictive scaling at lower compute scales is an important consideration. This recommendation aligns with the concurrent work by Gu et al. (2024).

More broadly, if sufficient task relevant data is available, then training on the test task can result in much more predictable scaling by shifting emergence to smaller compute scales. That is, scaling laws where models across scales are fine-tuned on the same, sufficient task-relevant data before evaluation. Such scaling laws correspond to those of specialist models, which for some tasks –e.g., legal annotation (Dominguez-Olmedo et al., 2024)– or purposes –e.g., safety– might be preferable to the scaling laws of generalist models.

# 6    DISCUSSION

The 1968 Olympics took place in Mexico City at the significant altitude of 2340 meters, higher than Australia's tallest peak. Runners who had trained at altitude in their home countries were better prepared to compete in Mexico City's conditions, as it turned out. But the hotly debated results of the Games did not lead the organizers to prohibit training at natural altitude. Instead, they let everyone do it, and athletes came to consider altitude training an excellent way to train.

The anecdote holds a lesson for the evaluation of large language models half a century later. Knowledge about the evaluation conditions necessarily influences training practices under competitive pressure. It may be a fool's errand to prohibit the practice. Instead, we propose to adjust for it by giving every model the same task-specific preparation before evaluation. We work from the assumption that training on the test task, in general, cannot be effectively detected, disallowed, or disincentivized. Detecting what training data a model has seen is a notoriously difficult problem –existing heuristics achieve partial success at best. Researchers routinely acknowledge the futility of fighting data contamination. Moreover, we anticipate that the ways to effectively train on the test task will only grow in scope and adoption.

Our work demonstrates that comparisons of different models are confounded by the choice of training data and training practices. Different model families vary in the degree that they were—implicitly or explicitly—trained on various test tasks. A relatively small amount of task data can have a disproportionally large effect on benchmark performance. It therefore makes little sense to compare model performance at face value without accounting for how the training data relate to the test task.

Training on the test task also has profound implications for the study of emergent behavior. After training on the test task, model capabilities become predictable at smaller scales. This greatly reduces the unpredictability associated with emergence, notably without any change in the metric.

Despite the daunting challenges that training on the test task poses for the fair evaluation of language models, it's also its own best remedy. Giving each model the same sufficient task-specific fine-tuning harmonizes model comparisons and linearizes the relationship between model capabilities and pretraining compute. We hope that our work informs stronger evaluation standards that address central challenges in the current evaluation ecosystem.

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

## A RELATED WORK

Benchmarks have played a central role in both machine learning (Hardt & Recht, 2022) and natural language processing (Storks et al., 2019). Classically, benchmarks comprised both a test set and a reasonably large training set (Garofolo et al., 1993; LeCun et al., 1998; Sang & De Meulder, 2003; Koehn, 2005; Deng et al., 2009). Models were trained on the same training set, and then evaluated on the accompanying test set. The success of unsupervised language modelling (Peters et al., 2018; Kenton & Toutanova, 2019; Radford et al., 2019), however, has changed this paradigm. Firstly, present-day language models differ in their training data, which is not standardized but rather treated as a design choice (Raffel et al., 2020; Albalak et al., 2024; Li et al., 2024). Secondly, language models are a priori not trained with the explicit objective of maximizing any single benchmark.

**Data contamination.** Data contamination or test-set contamination refers to any overlap between the training and the test data such that test results overestimate a model's generalization performance. The scale and often little curation of present-day pretraining corpora exacerbates data contamination concerns in language model evaluations (Jiang et al., 2024). Consequently, data contamination is usually discussed in the technical reports accompanying model releases (Radford et al., 2019; Brown et al., 2020; Chowdhery et al., 2023; Touvron et al., 2023b). However, detecting and preventing data contamination is currently an open problem (Gunasekar et al., 2023; Yang et al., 2023b; Golchin & Surdeanu, 2023). Roberts et al. (2023) and Li & Flanigan (2024) find that models often perform better on datasets that were publicly available during model training. While almost all models that we consider were released after MMLU and GSM8K, we nonetheless find that, controlling for compute, more recent models perform better. These performance gains are unlikely to be driven solely by test set leakage and require additional explanation. In Section 3.2, we find evidence that that training on the test task may be a more dominant factor in benchmark performance than data contamination. A key insight of our work is that models can train on the test task (e.g., multiple-choice question answering) and do much better at a given benchmark (e.g., MMLU) without necessarily seeing any benchmark data. This defies traditional notions of data contamination, which strictly refer to the leakage of benchmark data (Magar & Schwartz, 2022; Sainz et al., 2023; Dong et al., 2024). Moreover, whereas the core concern with data leakage is that benchmarks may *inadvertently* find their way to the training data, training on the test task is often a deliberate design choice (e.g., pre-training on instruction data).

**Adaptation prior to evaluation.** In the 2010s, language models had to be adapted to different benchmarks using supervised task data (Collobert et al., 2011; Dai & Le, 2015; Devlin et al., 2019). The purpose of such fine-tuned *benchmark models* was solely to facilitate the relative comparison of base models. Importantly, all models were adapted using the same supervised data (Collobert et al., 2011). With GPT-3 (Brown et al., 2020), few-shot prompting emerged as the dominant paradigm for adapting models to a particular task prior to evaluation (Liang et al., 2023), arguably due to its simplicity relative to fine-tuning. Bommasani et al. (2021) argue that benchmark evaluations should account for the adaptation resources used by each model (e.g., the adaptation data). Similarly, Liang et al. (2023) argue that the strategy for adapting the models to the benchmark evaluation should be controlled for. Clearly, few-shot prompting is a weaker form of adaptation than training on hundreds of thousands of task examples, as newer models often do. Our proposal of fine-tuning on the same, sufficient amount of task-specific data prior to evaluation aims to effectively control for model adaptation, ensuring that all models are given equal adaptation resources.

**Training on the test task.** The effectiveness of fine-tuning on the training set accompanying LLM benchmarks is well-known (Wei et al., 2022a; Wang et al., 2022; Chung et al., 2024). Consequently, many influential instruction-tuning datasets contain or are partly derived from benchmark train data (Wei et al., 2022a; Honovich et al., 2022; Mukherjee et al., 2023). Li & Flanigan (2024) identify small amounts of benchmark-specific data in the publicly available Alpaca (Taori et al., 2023) and Vicuna (Chiang et al., 2023) instruction-tuning sets. Zhou et al. (2023b) empirically analyze the effects of fine-tuning on benchmark-specific data and warn about its impacts on benchmark validity. To circumvent these issues, recent work has focused on indirect indicators of broader data contamination, such as a lack of robustness to task transformations (Wu et al., 2023), or underperformance on benchmarks with novel task combinations (Yu et al., 2023a). In contrast, we find evidence for training on the test task without the need for explicitly identifying specific data

points used at training time, or modifying tasks. In addition, our proposed method of fine-tuning on task data before evaluation allows us to quantify and correct for the effects of training on the test task on benchmark performance.

**Emergent abilities of language models.** Emergent capabilities (Wei et al., 2022b; Ganguli et al., 2022) refer to levels of model performance at large scales that cannot be easily predicted by extrapolating from smaller scales. Wei et al. (2022b) report emergent capabilities for various benchmarks including MMLU and GSM8K (Srivastava et al., 2022). However, Srivastava et al. (2022); Schaeffer et al. (2024b) find that the log-probability of the correct answer often improves smoothly, even when other metrics seem to show emergence. Rogers & Luccioni (2024) question the dominant definition of emergence and emphasize the importance of relating the training data to the test data before making claims about emergence. Lu et al. (2023) argue that most emergent capabilities can be explained by in-context-learning. Gadre et al. (2024) find that a model's perplexity on its pre-training data reliably predicts its average downstream performance. However, their analysis does not include the MMLU and GSM8K benchmarks, which are central to our work. Schaeffer et al. (2024a) argue that emergent capabilities are mostly an artifact of non-linear and discontinuous evaluation metrics like accuracy. In contrast, we find signs of emergence on MMLU even when using continuous metrics like the Brier score. Similarly to our findings, Snell et al. (2024) show that increasingly fine-tuning on the test task shifts the point of emergence to smaller compute scales.

# B  ADDITIONAL EXPERIMENTAL DETAILS

## B.1  MODELS CONSIDERED

Model size in billions of parameters is indicated by $N$ and pretraining data size in trillions of tokens is indicated by $D$. Model weights were retrieved from the corresponding HuggingFace (HF) repositories.

| Name | Train date | N | D | HF repository | Citation |
|---|---|---|---|---|---|
| baichuan-13b | 2023-06 | 13 | 1.4 | baichuan-inc/Baichuan-13B-Base | Yang et al. (2023a) |
| baichuan-7b | 2023-06 | 7 | 1.2 | baichuan-inc/Baichuan2-7B-Base | Yang et al. (2023a) |
| baichuan2-13b | 2023-09 | 13 | 2.6 | baichuan-inc/Baichuan2-13B-Base | Yang et al. (2023a) |
| baichuan2-7b | 2023-09 | 7 | 2.6 | baichuan-inc/Baichuan2-7B-Base | Yang et al. (2023a) |
| falcon-11b | 2024-05 | 11 | 5.0 | tiiuae/falcon-11B | Almazrouei et al. (2023) |
| falcon-7b | 2023-04 | 7 | 1.5 | tiiuae/falcon-7b | Almazrouei et al. (2023) |
| gemma-2b | 2024-02 | 2 | 3.0 | google/gemma-2b | Gemma et al. (2024) |
| gemma-7b | 2024-02 | 7 | 6.0 | google/gemma-7b | Gemma et al. (2024) |
| gpt-j-6b | 2021-03 | 6 | 0.4 | EleutherAI/gpt-j-6b | Wang & Komatsuzaki (2021) |
| internlm-20b | 2023-09 | 20 | 2.3 | internlm/internlm-20b | InternLM (2023) |
| internlm-7b | 2023-07 | 7 | 1.0 | internlm/internlm-7b | InternLM (2023) |
| internlm2-base-20b | 2024-01 | 20 | 2.6 | internlm/internlm2-base-20b | Cai et al. (2024) |
| internlm2-base-7b | 2024-01 | 7 | 2.6 | internlm/internlm2-base-7b | Cai et al. (2024) |
| llama-13b | 2023-02 | 13 | 1.0 | None | Touvron et al. (2023a) |
| llama-2-13b | 2023-07 | 13 | 2.0 | meta-llama/Llama-2-13b-hf | Touvron et al. (2023b) |
| llama-2-70b | 2023-07 | 70 | 2.0 | meta-llama/Llama-2-70b-hf | Touvron et al. (2023b) |
| llama-2-7b | 2023-07 | 7 | 2.0 | meta-llama/Llama-2-7b-hf | Touvron et al. (2023b) |
| llama-3-8b | 2024-04 | 8 | 15.0 | meta-llama/Meta-Llama-3-8B | MetaAI (2024) |
| llama-30b | 2023-02 | 32.5 | 1.4 | None | Touvron et al. (2023a) |
| llama-65b | 2023-02 | 65.2 | 1.4 | None | Touvron et al. (2023a) |
| llama-7b | 2023-02 | 7 | 1.0 | None | Touvron et al. (2023a) |

| map-neo-7b | 2024-05 | 7 | 4.5 | m-a-p/neo_7b | Zhang et al. (2024) |
|---|---|---|---|---|---|
| olmo-1.7-7b | 2024-04 | 7 | 2.0 | allenai/OLMo-1.7-7B-hf | Groeneveld et al. (2024) |
| olmo-1b | 2024-01 | 1 | 2.0 | allenai/OLMo-1B-hf | Groeneveld et al. (2024) |
| olmo-7b | 2024-01 | 7 | 2.5 | allenai/OLMo-7B-hf | Groeneveld et al. (2024) |
| openllama-13b | 2023-06 | 13 | 1.0 | openlm-research/open_llama_13b | OpenLlama (2023) |
| openllama-3b | 2023-06 | 3 | 1.0 | openlm-research/open_llama_3b | OpenLlama (2023) |
| openllama-3b-v2 | 2023-07 | 3 | 1.0 | openlm-research/open_llama_3b_v2 | OpenLlama (2023) |
| openllama-7b | 2023-06 | 7 | 1.0 | openlm-research/open_llama_7b | OpenLlama (2023) |
| openllama-7b-v2 | 2023-07 | 7 | 1.0 | openlm-research/open_llama_7b_v2 | OpenLlama (2023) |
| pythia-1.4b | 2022-10 | 1.4 | 0.3 | EleutherAI/pythia-1.4b | Biderman et al. (2023) |
| pythia-12b | 2022-10 | 12 | 0.3 | EleutherAI/pythia-12b | Biderman et al. (2023) |
| pythia-160m | 2022-10 | 0.16 | 0.3 | EleutherAI/pythia-160m | Biderman et al. (2023) |
| pythia-1b | 2022-10 | 1 | 0.3 | EleutherAI/pythia-1b | Biderman et al. (2023) |
| pythia-2.8b | 2022-10 | 2.8 | 0.3 | EleutherAI/pythia-2.8b | Biderman et al. (2023) |
| pythia-410m | 2022-10 | 0.41 | 0.3 | EleutherAI/pythia-410m | Biderman et al. (2023) |
| pythia-6.9b | 2022-10 | 6.9 | 0.3 | EleutherAI/pythia-6.9b | Biderman et al. (2023) |
| pythia-70m | 2022-10 | 0.07 | 0.3 | EleutherAI/pythia-70m | Biderman et al. (2023) |
| qwen-1.5-0.5b | 2024-01 | 0.5 | 2.4 | Qwen/Qwen1.5-0.5B | Bai et al. (2023) |
| qwen-1.5-1.8b | 2024-01 | 1.8 | 2.4 | Qwen/Qwen1.5-1.8B | Bai et al. (2023) |
| qwen-1.5-14b | 2024-01 | 14 | 4.0 | Qwen/Qwen1.5-14B | Bai et al. (2023) |
| qwen-1.5-4b | 2024-01 | 4 | 2.4 | Qwen/Qwen1.5-4B | Bai et al. (2023) |
| qwen-1.5-7b | 2024-01 | 7 | 4.0 | Qwen/Qwen1.5-7B | Bai et al. (2023) |
| qwen2-0.5b | 2024-06 | 0.5 | 12.0 | Qwen/Qwen2-0.5B | Yang et al. (2024) |
| qwen2-1.5b | 2024-06 | 1.5 | 7.0 | Qwen/Qwen2-1.5B | Yang et al. (2024) |
| qwen2-7b | 2024-06 | 7 | 7.0 | Qwen/Qwen2-7B | Yang et al. (2024) |
| redpajama-3b | 2023-05 | 3 | 0.8 | togethercomputer/RedPajama-INCITE-Base-3B-v1 | TogetherWeCompute (2023) |
| redpajama-7b | 2023-05 | 7 | 1.0 | togethercomputer/RedPajama-INCITE-7B-Base | TogetherWeCompute (2023) |
| skywork-13b | 2023-10 | 13 | 3.2 | Skywork/Skywork-13B-base | Wei et al. (2023) |
| stablelm-2-1.6b | 2024-01 | 1.6 | 2.0 | stabilityai/stablelm-2-1_6b | Bellagente et al. (2024) |
| stablelm-2-12b | 2024-03 | 12.1 | 2.0 | stabilityai/stablelm-2-12b | Bellagente et al. (2024) |
| stablelm-3b-4e1t | 2023-09 | 2.8 | 4.0 | stabilityai/stablelm-3b-4e1t | StabilityAI (2023) |
| stablelm-base-alpha-3b-v2 | 2023-08 | 2.8 | 1.1 | stabilityai/stablelm-base-alpha-3b-v2 | StabilityAI (2023) |
| stablelm-base-alpha-7b-v2 | 2023-08 | 7 | 1.1 | stabilityai/stablelm-base-alpha-7b-v2 | StabilityAI (2023) |
| yi-6b | 2023-11 | 6 | 3.1 | 01-ai/Yi-1.5-6B | Young et al. (2024) |
| ziya2-13b-base | 2023-11 | 13 | 2.65 | IDEA-CCNL/Ziya2-13B-Base | Gan et al. (2023) |

### B.1.1 INSTRUCTION-TUNED AND CHAT MODELS

| Name | Train date | Base model | HF repository | Citation |
|---|---|---|---|---|
| falcon-7b-instruct | 2023-04 | falcon-7b | tiiuae/falcon-7b-instruct | Almazrouei et al. (2023) |
| gemma-2b-instruct | 2024-02 | gemma-2b | google/gemma-2b-it | Gemma et al. (2024) |
| gemma-7b-instruct | 2024-02 | gemma-7b | google/gemma-7b-it | Gemma et al. (2024) |
| internlm-chat-20b | 2023-09 | internlm-20b | internlm/internlm-chat-20b | InternLM (2023) |
| internlm-chat-7b | 2023-07 | internlm-7b | internlm/internlm-chat-7b | InternLM (2023) |

| | | | | |
|---|---|---|---|---|
| internlm2-7b | 2024-01 | internlm2-base-7b | internlm/internlm2-7b | Cai et al. (2024) |
| internlm2-chat-1_8b | 2024-01 | internlm2-base-7b | internlm/internlm2-chat-1_8b | Cai et al. (2024) |
| internlm2-chat-20b | 2024-01 | internlm2-base-20b | internlm/internlm2-chat-20b | Cai et al. (2024) |
| internlm2-chat-7b | 2024-01 | internlm2-base-7b | internlm/internlm2-chat-7b | Cai et al. (2024) |
| llama-2-13b-chat | 2023-07 | llama-2-13b | meta-llama/Llama-2-13b-chat-hf | Touvron et al. (2023b) |
| llama-2-7b-chat | 2023-07 | llama-2-7b | meta-llama/Llama-2-7b-chat-hf | Touvron et al. (2023b) |
| llama-3-8b-instruct | 2024-04 | llama-3-8b | meta-llama/Meta-Llama-3-8B-Instruct | MetaAI (2024) |
| map-neo-7b-instruct | 2024-05 | map-neo-7b | m-a-p/neo_7b_instruct_v0.1 | Zhang et al. (2024) |
| map-neo-7b-sft | 2024-05 | map-neo-7b | m-a-p/neo_7b_sft_v0.1 | Zhang et al. (2024) |
| olmo-7b-0724-instruct-hf | 2024-01 | olmo-7b | allenai/OLMo-7B-0724-Instruct-hf | Groeneveld et al. (2024) |
| olmo-7b-0724-sft-hf | 2024-01 | olmo-7b | allenai/OLMo-7B-0724-SFT-hf | Groeneveld et al. (2024) |
| olmo-7b-instruct-hf | 2024-01 | olmo-7b | allenai/OLMo-7B-Instruct-hf | Groeneveld et al. (2024) |
| olmo-7b-sft-hf | 2024-01 | olmo-7b | allenai/OLMo-7B-SFT-hf | Groeneveld et al. (2024) |
| qwen-1.5-0.5b-chat | 2024-01 | qwen-1.5-0.5b | Qwen/Qwen1.5-0.5B-Chat | Bai et al. (2023) |
| qwen-1.5-1.8b-chat | 2024-01 | qwen-1.5-1.8b | Qwen/Qwen1.5-1.8B-Chat | Bai et al. (2023) |
| qwen-1.5-14b-chat | 2024-01 | qwen-1.5-14b | Qwen/Qwen1.5-14B-Chat | Bai et al. (2023) |
| qwen-1.5-4b-chat | 2024-01 | qwen-1.5-4b | Qwen/Qwen1.5-4B-Chat | Bai et al. (2023) |
| qwen-1.5-7b-chat | 2024-01 | qwen-1.5-7b | Qwen/Qwen1.5-7B-Chat | Bai et al. (2023) |
| redpajama-7b-chat | 2023-05 | redpajama-7b | togethercomputer/RedPajama-INCITE-7B-Chat | TogetherWeCompute (2023) |
| redpajama-chat-3b-v1 | 2023-05 | redpajama-3b | togethercomputer/RedPajama-INCITE-Chat-3B-v1 | TogetherWeCompute (2023) |
| redpajama-instruct-3b-v1 | 2023-05 | redpajama-3b | togethercomputer/RedPajama-INCITE-Instruct-3B-v1 | TogetherWeCompute (2023) |
| redpajama-instruct-7b | 2023-05 | redpajama-7b | togethercomputer/RedPajama-INCITE-7B-Instruct | TogetherWeCompute (2023) |
| stablelm-2-1.6b-chat | 2024-01 | stablelm-2-1.6b | stabilityai/stablelm-2-1_6b-chat | Bellagente et al. (2024) |
| stablelm-2-12b-chat | 2024-03 | stablelm-2-12b | stabilityai/stablelm-2-12b-chat | Bellagente et al. (2024) |
| stablelm-2-zephyr-1.6b | 2024-06 | stablelm-2-1.6b | stabilityai/stablelm-2-zephyr-1_6b | Bellagente et al. (2024) |
| stablelm-zephyr-3b | 2023-11 | stablelm-3b-4e1t | stabilityai/stablelm-zephyr-3b | Tunstall et al. (2023) |
| vicuna-13b-v1.1 | 2023-04 | llama-13b | lmsys/vicuna-13b-v1.1 | Chiang et al. (2023) |
| vicuna-13b-v1.3 | 2023-06 | llama-13b | lmsys/vicuna-13b-v1.3 | Chiang et al. (2023) |
| vicuna-7b-v1.1 | 2023-04 | llama-7b | lmsys/vicuna-7b-v1.1 | Chiang et al. (2023) |
| vicuna-7b-v1.3 | 2023-06 | llama-7b | lmsys/vicuna-7b-v1.3 | Chiang et al. (2023) |

## B.2 FINE-TUNING HYPERPARAMETERS

We fine-tune all model parameters. For models with less than 10B parameters, we fine-tune on a single GPU with BF16 precision. For models between 10B and 30B parameters, we train on a single H100 node using DeepSpeed ZeRO-3 (Rajbhandari et al., 2020) and full precision. For models with more than 30B parameters, we train on two H100 nodes using DeepSpeed ZeRO-3 and full precision. Due to the large compute cost of the

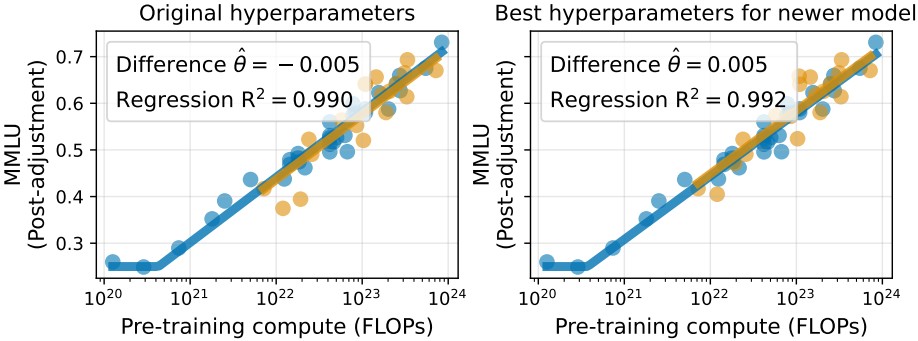

Figure 9: We perform a learning rate sweep using values $[6 \cdot 10^{-5}, 2 \cdot 10^{-5}, 6 \cdot 10^{-6}, 2 \cdot 10^{-6}, 6 \cdot 10^{-7}]$. The left plot shows MMLU performance after fine-tuning with the original learning rates detailed in Appendix B.2. The right plot shows MMLU performance after fine-tuning but using for newer models the optimal learning rate from the sweep that maximizes MMLU performance. Even with the advantage of a higher hyperparameter search budget for the newer models, the estimated effect size $\hat{\theta}$ of model recency on benchmark performance remains both small and not statistically significant.

experiments, we perform minimal hyperparameter tuning and use standard hyperparameter choices throughout. We use a learning rate of $2 \cdot 10^{-5}$ for models with fewer than 10B parameters and a learning rate of $2 \cdot 10^{-6}$ for models with more than 10B parameters. For Qwen 2 as well as four of the 7B models –Gemma 7B, Olmo 7B, Olmo 1.7 7B, and Llama 3 8B– benchmark accuracy degraded after fine-tuning. For these models, we use a peak learning rate of $2 \cdot 10^{-6}$ instead. We use a cosine learning rate schedule with linear warm-up for 50 steps and decay to $10\%$ of the peak learning rate. We use AdamW (Loshchilov & Hutter, 2018) as the optimizer, with $\beta_1 = 0.9$, $\beta_2 = 0.95$, and $\epsilon = 10^{-8}$. We fine-tune with batch size 64. We use a weight decay rate of 0.1 and clip gradients at 1.0. We verify that the training loss decreases for all models on both of the fine-tuning datasets. To reduce the computation burden of fine-tuning, we train with context size 600. We verify that less than $5\%$ of the fine-tuning examples have context length above 600.

We use an internal cluster of A100 and H100 GPUs. Fine-tuning all models required approximately 10,000 H100 GPU hours, whereas evaluating all models in the different benchmarks required approximately 400 H100 GPU hours.

### B.3 ROBUSTNESS CHECK ON THE HYPERPARAMETER SEARCH BUDGET

We found the learning rate to be, by a large margin, the single most impactful hyperparameter. We perform a sweep on the MMLU auxiliary training set with the following learning rates: $[6 \cdot 10^{-5}, 2 \cdot 10^{-5}, 6 \cdot 10^{-6}, 2 \cdot 10^{-6}, 6 \cdot 10^{-7}]$. We are unable to perform more extensive hyperparameter sweeps due to their large computational cost. We find that no models benefit from a smaller learning rate than $2 \cdot 10^{-6}$, and only one model benefits from a larger learning rate than $2 \cdot 10^{-5}$, Pythia 70M (the model with smallest pre-training compute). Thus, for every model except for Pythia 70M, the optimal learning rate is inside the boundary of the hyperparameter sweep.

A potential concern is that older models may appear to match the performance of newer models because the fine-tuning hyperparameters could be systematically more favorable to the older models. To address this concern, we recreate our main experiment by fine-tuning older models with their original learning rates, while selecting for the newer models the optimal learning rate in the sweep that leads to highest MMLU performance. That is, we deliberately give newer models a systematic advantage in terms of hyperparameter search budget. We plot the results in Figure 9. The estimated effect size of model recency on benchmark performance shifts from $\hat{\theta} = -0.005$ to $\hat{\theta} = 0.005$. Therefore, despite giving newer models a systemic advantage in terms of hyperparameter search budget, the effect size remains both remains small and not statistically significant.

## C CAUSAL INTERPRETATION OF OUR FINDINGS

In Section 2.2 we established that models trained after November 2023 significantly outperform those trained before November 2023 for both MMLU and GSM8K. We then showed that fine-tuning all models in the test task equalizes the performance of newer and older models. We now present a causal interpretation of our

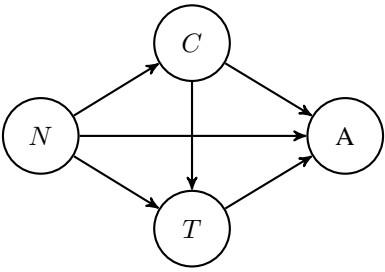

Figure 10: Whether a model was trained after November 2023 ($N$) influences its pretraining compute ($C$) and how much it trains on the test task ($T$). All three influence the benchmark accuracy ($A$) of the model.

findings, aiming to quantify the extent to which the effect of model recency $N$ on benchmark accuracy $A$ is mediated by training on the test task $T$.

The key obstacle to our analysis is that test task training $T$ is unobservable. Firstly, because practitioners are typically not transparent about their design choices, including the pretraining data. Secondly, because the extent to which different training practices might amount to test task training is unclear. Nonetheless, by fine-tuning on task-specific data, we can intervene on the extent to which models train on the test task.

Figure 10 summarizes our causal assumption. The time at which a model was trained determines the design choices made, such as its pretraining data or pretraining compute $C$. These design choices in turn affect how much the model trains on the test task. All these factors ultimately influence the pretrained model and thus its benchmark performance. We also admit that compute might influence test task training. For instance, pre-training on larger datasets may lead to models training more on the test task.

We interpret the proposed adjustment method as intervening on the test task training variable $T$. Namely, by fine-tuning all models on the same amount of task-specific data before evaluation. The external validity of our subsequent analysis hinges on the assumption that our controlled experimental setting –fine-tuning models after the pretraining stage– is reasonably similar to the natural settings in which practitioners might train on the test task during pretraining (e.g., by including instruction data in the pretraining data mixture). We provide evidence for this in Appendix D.3.

We model fine-tuning as a hard intervention $\mathrm{do}(T = t)$ (Pearl, 2009). The specific magnitude of the intervention $t$ need not be quantified. Instead, the key assumption is that by fine-tuning on the same, sufficient amount of task data, all models will have received the same amount of test task training. Since some base models may have already trained on the test task before fine-tuning, this assumption only holds if test task training saturates, and we train on enough task data to reach saturation. The fact that our task-specific datasets allow older models to match the performance of newer models provides some evidence that we train on enough task-specific data to reach saturation.

We draw inspiration from scaling laws (Kaplan et al., 2020) and model the relationship between pretraining compute and its causal descendants as piecewise log-linear:

$$f(C, \alpha) = \alpha_0 + \sum_{i=1}^{|\alpha|} \alpha_i \log C \cdot [C > c_i] \tag{2}$$

For simplicity, we consider three fixed knots at $c_1 = 0$, $c_2 = 10^{22}$, and $c_3 = 10^{23}$ FLOPs. We assume all other variable relationships to be linear, resulting in the structural assignments:

$$T := f(C, \beta) + \phi N + \delta, \quad \delta \sim \mathcal{N}(0, \sigma_\delta^2) \tag{3}$$
$$A := f(C, \alpha) + \psi N + \gamma T + \eta + \epsilon, \quad \epsilon \sim \mathcal{N}(0, \sigma_\epsilon^2) \tag{4}$$

We denote benchmark accuracy after fine-tuning as $A|_{\mathrm{do}(T=t)}$. To estimate the direct effect $N \to A$ of model recency on accuracy, we regress the linear model

$$
\begin{aligned}
A|_{\mathrm{do}(T=t)} &= f(C, \alpha) + \psi N + \gamma t + \eta + \epsilon \\
&= f(C, \alpha) + \psi N + \eta' + \epsilon, \quad \eta' = \eta + \gamma t
\end{aligned}
\tag{5}
$$

where $\alpha, \psi, \eta'$ are the fit's parameters and $\epsilon$ is random noise. The coefficient $\psi$ corresponds to the direct effect $N \to A$ of model recency on benchmark accuracy. We additionally regress on the difference in accuracy pre-

Table 3: The indirect effect $N \rightarrow T \rightarrow A$ mediated by test task training $T$ is positive, significant, and large: newer models attain higher benchmark scores primarily because of training on the test task.

|  | MMLU | GSM8K |
|---|---|---|
| $\hat{\phi}$ | **0.071** (0.018) | **0.168** (0.032) |
| $R^2$ | 0.530 | 0.503 |

Standard errors in parentheses. Bold indicates $p < 0.05$.

Table 4: We find no evidence of a significant direct effect of model recency $N$ on accuracy $A$, that is, of the improvements of newer models being attributable to anything else other than training on the test task.

|  | MMLU | GSM8K |
|---|---|---|
| $\hat{\psi}$ | -0.004 (0.009) | 0.000 (0.032) |
| $R^2$ | 0.926 | 0.763 |

Standard errors in parentheses. Bold indicates $p < 0.05$.

and post-intervention

$$
\begin{aligned}
A - A|_{\text{do}(T=t)} &= (f(C, \alpha) + \psi N + \gamma T + \eta + \epsilon_1) - (f(C, \alpha) + \psi N + \gamma t + \eta + \epsilon_2) \\
&= \gamma T - \gamma t + \epsilon_1 - \epsilon_2 \\
&= f(C, \gamma\beta) + \gamma\phi N + \gamma\delta - \gamma t + \epsilon_1 - \epsilon_2 \\
&= f(C, \beta') + \phi' N + b + \epsilon', \quad \text{for } \beta' = \gamma\beta, \phi' = \gamma\phi, b = -\gamma t, \epsilon' = \epsilon_1 - \epsilon_2 + \gamma\delta
\end{aligned}
\tag{6}
$$

where $\beta'$, $\phi'$, $b$ are the fit's parameters and $\epsilon'$ is random noise. The coefficient $\phi'$ corresponds to the indirect effect $N \rightarrow T \rightarrow A$ of model recency $N$ on benchmark accuracy $A$ mediated by test task training $T$ (Pearl, 2013). That is, the improvements in accuracy of recent models attributable to training on the test task.

We fit the models in Equation 5 and Equation 6, and we report the coefficients pertaining to $N \rightarrow A$ and $N \rightarrow T \rightarrow A$ in Table 4 and Table 3. We find that the indirect effect $N \rightarrow T \rightarrow A$ of model recency on accuracy mediated by test task training $T$ is significant, positive, and large. In contrast, we find no evidence of a significant direct effect $N \rightarrow A$ of model recency on accuracy. We therefore find no evidence of the improvements of newer models being attributable to anything else other than training on the test task.

In conclusion, our causal analysis indicates that the differences in MMLU and GSM8K performance between newer and older models observed in Section 2.1 are largely attributable to differences in test task training. That is, the mechanism by which newer models outperform older models is primarily by training more on the test task.

# D ROBUSTNESS CHECK ON THE TEMPORAL SPLIT

## D.1 ADJUSTING THE TEMPORAL CUTOFF BY A FEW MONTHS

We repeat the analysis of Section 2 for two additional temporal splits: September 2023 and January 2024, and present the results in Figure 11 and Figure 12, respectively. Our results are robust to shifting the temporal cutoff by a few months. That is, our findings indicate that practitioners started adopting design choices around late 2023 that amount to models training on the test task much more, which is consistent with models' technical reports starting to mention the use of benchmark or instruction data at pre-training time. Choosing specifically the month of November as the cut-off is therefore not critical for our analysis.

## D.2 EN VS CN LANGUAGE DATA

Instead of diving models using a temporal split, we divide models based on whether they were trained primarily on English (EN) data or on a mixture of English and Chinese (EN+CN) language data. While there is a considerable overlap between the temporal split and the EN/EN+CN model split, there are notable differences. In particular, the Baichuan, Baichuan 2, and InternLM, and Skywork families were trained before November 2023 and trained on EN+CN data. Conversely, Gemma, Llama 3, StableLM 2, Falcon 2, and Olmo were trained after November 2023 and trained on EN data.

We repeat the analysis of Section 2 for the EN and EN+CN model split, see Figure 13. We observe that, controlling for pretraining compute, models trained on EN+CN language data outperform those trained primarily on EN by 9 accuracy points on MMLU and 12 accuracy points on GSM8K. After the proposed adjustment, however, the difference in performance between models trained on EN data and EN+CN data is small and not statistically significant.

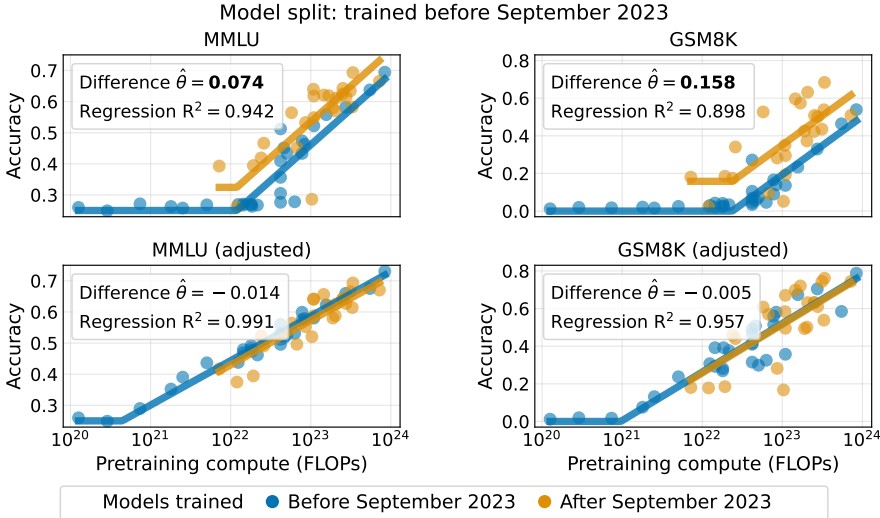

Figure 11: Robustness check with September 2023 as the temporal cutoff.

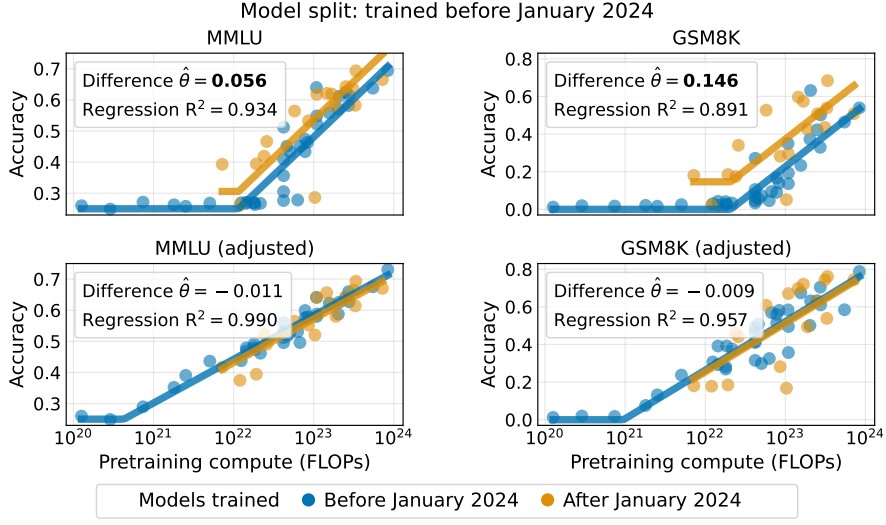

Figure 12: Robustness check with January 2024 as the temporal cutoff.

The confounding and measured effect sizes for the EN and EN+CN model split resemble those obtained for the temporal split, which we interpret as a valuable robustness check of our results.

### D.3 HOW SIMILAR ARE NEWER MODELS TO OLDER, FINE-TUNED MODELS?

In Section 3.1 we fine-tune older models on the test task, and we demonstrate that the differences in benchmark performance between the fine-tuned and non fine-tuned models resemble those between newer and older models. In this section we provide further evidence that newer models resemble older, fine-tuned models.

We take the older models and we fine-tune them with 64,000 training examples from the auxiliary training sets introduced in Section 2.1. We plot in Figure 14 the benchmark scores of the older, fine-tuned models as well as that of the newer models. We qualitatively observe that both the older, fine-tuned models and the newer models exhibit similar scaling. That is, older fine-tuned models resemble newer models in terms of performance per compute.

We perform a quantitative analysis consisting in discriminating between the older models and the newer models based on their pretraining compute and benchmark accuracy. That is, we construct a tabular dataset where rows

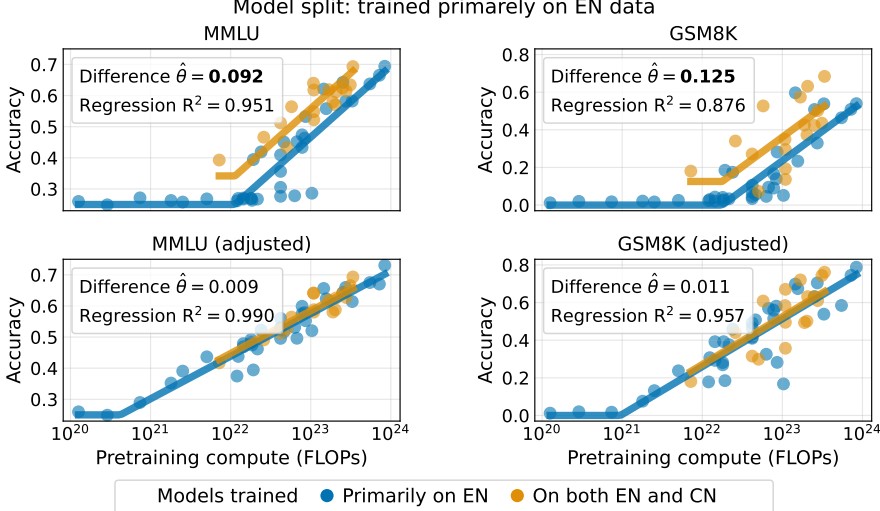

Figure 13: Models trained on both English (EN) and Chinese (CN) language data outperform those trained primarily on English data. After adjusting for test task training, we find no evidence of a significant difference $\theta$ in performance between models trained on EN data and EN+CN data.

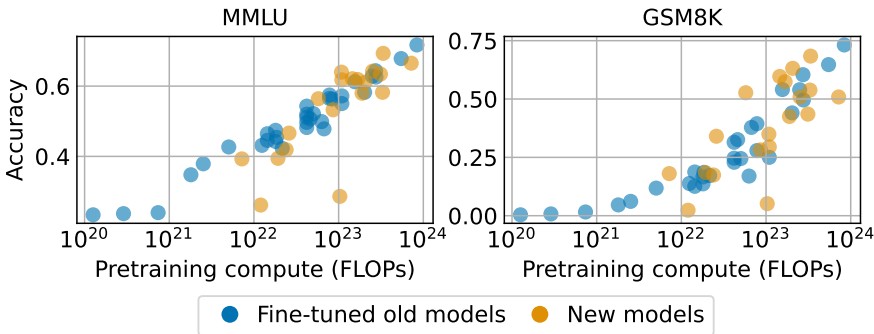

Figure 14: New models resemble old models that were fine-tuned. Temporal cut-off: November 2023.

Table 5: Accuracy in discriminating between older and newer models in terms of their pretraining compute and benchmark accuracy. Older, fine-tuned models are indistinguishable from newer models.

| Discriminator test | MMLU | GSM8K |
|---|---|---|
| Older models vs newer models | 64.6% | 73.9% |
| Fine-tuned, older models vs newer models | 52.2% | 52.5% |

Random chance accuracy is 50%.

are models and columns are their corresponding pretraining compute, benchmark accuracy, and whether the model was trained after November 2023. We then train a classifier aiming to predict model recency from compute and accuracy. Intuitively, if the performance of older models is very different from that of newer models, then we would obtain high prediction accuracy (i.e., the two classes are highly separable). Note that prediction accuracy provides a lower bound on the total variation (TV) distance between the distributions of compute and accuracy of older and newer models.

We train XGBoost classifiers and report balanced accuracy for leave-one-out cross-validation in Table 5. We obtain close to random-chance accuracy in discriminating between older, fine-tuned models and newer models. That is, older fine-tuned models are indistinguishable from newer models in terms of their performance per pre-training compute.

# E    RESULTS FOR THE OPENLLM LEADERBOARD v2

HuggingFace released on June 2024 a revision of the OpenLLM Leaderboard (Fourrier et al., 2024a). The HF leaderboard v2 differs from v1 in the six benchmarks it considers: MMLU Pro (Wang et al., 2024), GPQA (Rein et al., 2023), BBH (Suzgun et al., 2023), MuSR (Sprague et al., 2023), the Level 5 subset of MATH (Hendrycks et al., 2021), and IFEval (Zhou et al., 2023a). MMLU and GPQA test for knowledge and are framed as multiple-choice questions. BBH and MuSR test for reasoning. MATH tests for mathematical reasoning. IFEval tests the ability of models to follow instructions.

The creators of the OpenLLM Leaderboard cite contamination as a key motivation for releasing the v2 revision. They note that a key criterion in choosing the benchmarks of the HF leaderboard v2 was lack of contamination in models as of today. In particular, Fourrier et al. (2024b) claim that current models are not contaminated for GPQA, MuSR, and MMLU Pro: GPQA due to the gating of the test set, and MuSR and MMLU Pro due to their "youth". Fourrier et al. (2024b) succinctly express their concern as regards to data contamination in the HF leaderboard v1:

> *"Some newer models also showed signs of contamination. By this, we mean that models were possibly trained on benchmark data or on data very similar to benchmark data. As such, some scores stopped reflecting the general performance of the model and started to overfit on some evaluation datasets instead of reflecting the more general performance of the task being tested. This was, in particular, the case for GSM8K and TruthfulQA, which were included in some instruction fine-tuning sets."*

Note that *"models were possibly trained on benchmark data or on data very similar to benchmark data"* encompasses not only test set contamination but more broadly training on the test task.

We evaluate all models on MMLU Pro, GPQA, BBH, MuSR and MATH Lvl 5. We use the LM Evaluation Harness library in an identical fashion to the HF leaderboard v2. We do not evaluate on IFEval since it tests for instruction following and we evaluate base models. We additionally evaluate the models that we fine-tuned in Section 2.1 for multiple choice question answering and mathematical reasoning. This gives us models' adjusted benchmark scores after training on multiple choice question answering and mathematical reasoning. For MATH Lvl 5, we use the models fine-tuned on mathematical data, whereas for MMLU Pro, GPQA, BBH and MuSR we use the models fine-tuned on multiple choice question answering. The fine-tuning datasets were not adapted to the new benchmarks in the HF leaderboard v2, thus giving a valuable insight into how well these task-relevant datasets generalize beyond MMLU and GSM8K.

We plot in Figure 15 models benchmark scores pre- and post-post adjustment. We find that newer models significantly outperform older ones in all five benchmarks after controlling for pretraining compute. The differences in performance are smaller in absolute terms than those measured for MMLU (0.073) and GSM8K (0.191). This is in part because these benchmarks are "harder", meaning also smaller differences in performance between the best and worst model. For this reason, we also report the difference between newer and older models relative to the difference between the best and worst model. This relative difference is 13.7% for MMLU Pro, 14.5% for GPQA, 12.1% for MuSR, 9.7% for BBH, and 10.0% for MATH Lvl 5, compared to 15.3% for MMLU and 25.0% for GSM8K. Therefore, newer models overperform in MMLU Pro, GPQA and MuSR about as much as they do for MMLU, and somewhat less for BBH and MATH Lvl 5.

Fine-tuning on task-relevant data reduces the difference in performance between newer and older models for all five benchmarks. Therefore, we find evidence that training on the test task plays a substantial role in newer models outperforming older ones in the benchmarks of the HF Leaderboard v2. For GPQA and MuSR, the difference in performance after adjustment is very small ($|\hat{\theta}| \leq 0.002$) and not statistically significant. For BBH, the estimated difference in performance $\hat{\theta}$ reduces by 40% to 0.015 and is no longer statistically significant. For MMLU Pro and MATH Lvl 5 the difference reduces by 19% and 33% respectively but remains reasonably large ($\hat{\theta}$ ¿ 0.01).

One possible reason for the fact that the adjustment for MMLU Pro and MATH Lvl 5 is not as effective as for MMLU and GSM8K is that the fine-tuning examples are simply not as relevant for MMLU Pro and MATH Lvl 5. For example, the questions and answers in MATH Lvl 5 contain much more LaTeX equation formatting than our mathematical reasoning fine-tuning dataset. Similarly, our multiple choice fine-tuning dataset contains mostly questions with 4 answer choices, whereas all MMLU Pro questions have 10 answer choices. Thus, models are primarily fine-tuned to answer "A", "B", "C", and "D" but not "E", "F", "G". We modify MMLU Pro to contain questions with 4 answer choices by randomly discarding 6 of the incorrect answer choices. We

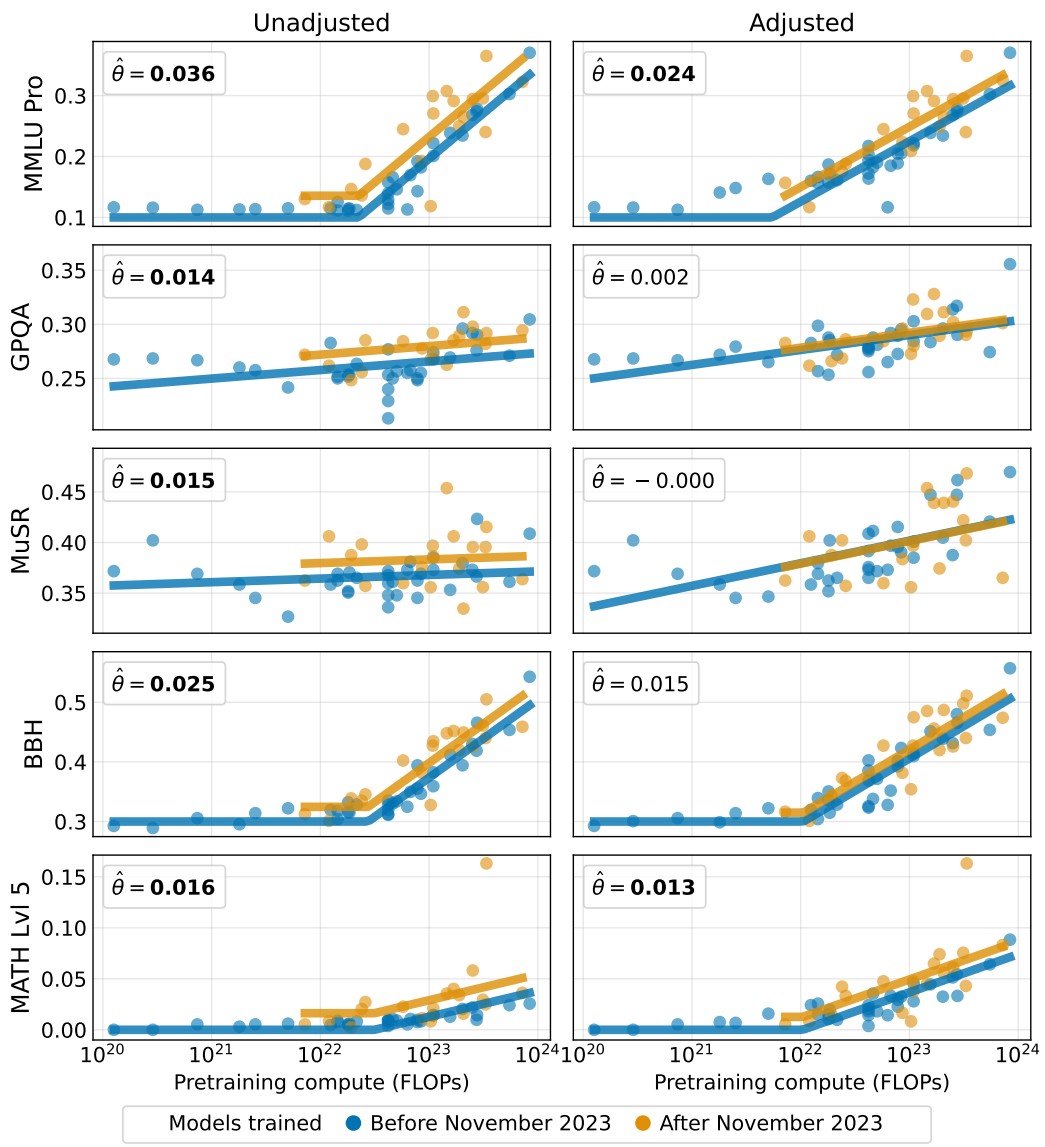

Bold indicates statistical significance with p ¡ 0.05.

Figure 15: Results for the OpenLLM Leaderboard v2. For all benchmarks, models trained after November 2023 significantly outperform models trained before November 2023 when controlling for pretraining compute. After fine-tuning models on multiple choice question answering and mathematical reasoning, differences in performance between newer and older models reduce for all five benchmarks. These differences are no longer significant for GPQA, MuSR and BBH, but remain significant for MMLU Pro and MATH Lvl 5.

evaluate models pre- and post-adjustment and plot the results in Figure 16. We observe that the difference in performance between newer and older models after adjustment reduces from 0.024 to 0.016, and is no longer statistically significant. This observation suggests that fine-tuning one more relevant task-data might further reduce the gap between newer and older models in MMLU Pro and MATH Lvl 5.

**Discussion.**    Fourrier et al. (2024b) cite newer models overperforming in the HF leaderboard v1 due to being "possibly trained on benchmark data or on data very similar to benchmark data" as a major reason for the HF leaderboard v2 revision. We, however, find evidence that training on the test task is also a confounder for the newly included benchmarks. Specifically, the difference in performance between newer and older models is

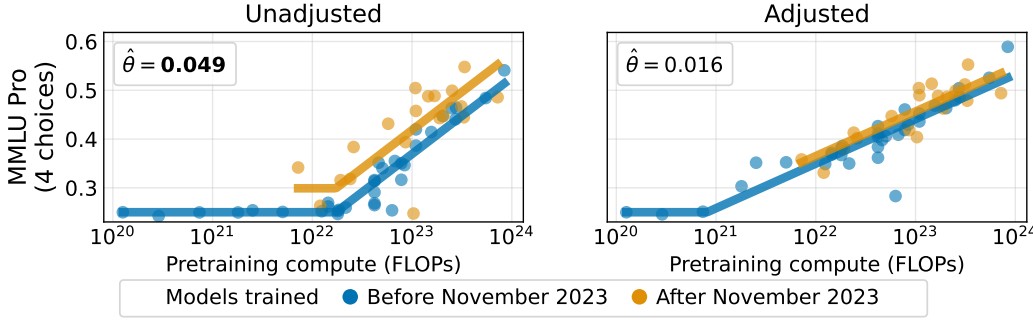

Bold indicates statistical significance with $p < 0.05$.

Figure 16: We modify MMLU Pro to only contain questions with 4 answer choices by for every question randomly discarding 6 of the incorrect answer choices. After adjustment, the difference in performance $\hat{\theta}$ between newer and older models is smaller and no longer statistically significant.

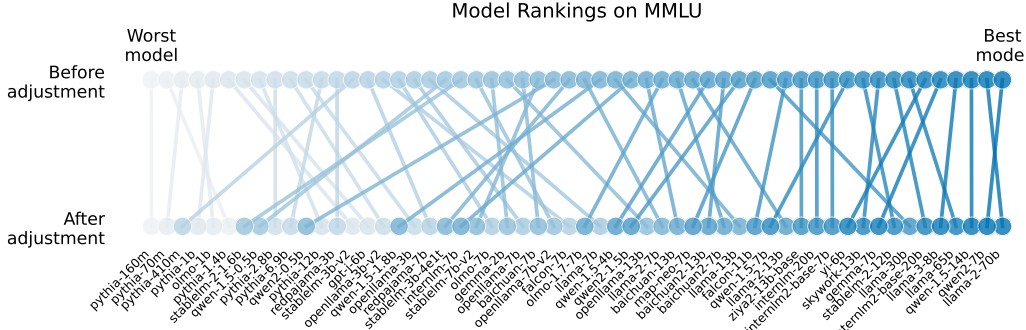

Figure 17: Training on the test task significantly alters model rankings on MMLU.

significant for MMLU Pro, GPQA, MuSR, BBH and MATH Lvl 5, and these differences reduce after adjusting by fine-tuning on the test task.

Fourrier et al. (2024b) explicitly highlight GPQA and MuSR as benchmarks likely unaffected by contamination, the former due to being gated and the latter due to its "youth". Not only do newer models significantly outperform older ones in GPQA and MuSR, but these differences in performance fully vanish after fine-tuning on the test task. That is, newer models likely overperform in GPQA and MuSR precisely due to training on the test task.

These findings highlight that training on the test task is a distinct phenomenon from test set leakage. Strategies that aim to mitigate data contamination –e.g., dynamic benchmarks– might not be effective in mitigating the confounding effect of training on the test task. In contrast, we extensively demonstrated the effectiveness of our proposed adjustment procedure, that is, fine-tuning on sufficient task-relevant data before evaluation.

## F    ADDITIONAL FIGURES

**MMLU rankings**    Training on the test task significantly alters model rankings on MMLU, with an average shift of 4.8 ranks and a maximum shift of 16 ranks.

**Reformulating ARC and HellaSwag as multiple choice**    In Figure 18 we show that ARC and HellaSwag do not exhibit emergence when using the standard cloze evaluation. When reformulating the task as multiple choice in the style of MMLU, however, we observe emergence around $10^{22}$ to $10^{23}$ FLOPs, similarly to MMLU. Emergence in this range of compute persists even when changing the evaluation metric from accuracy to Brier score –a continuous metric–, as suggested by Schaeffer et al. (2024a).

**Emergence for GSM8K as models train on the test task**    Similar to MMLU, we find that increasingly fine-tuning models on mathematical reasoning makes the phenomenon of emergence gradually disappear, see Figure 19. The point of emergence arises at increasingly lower scales, recovering cleaner log-linear fits.

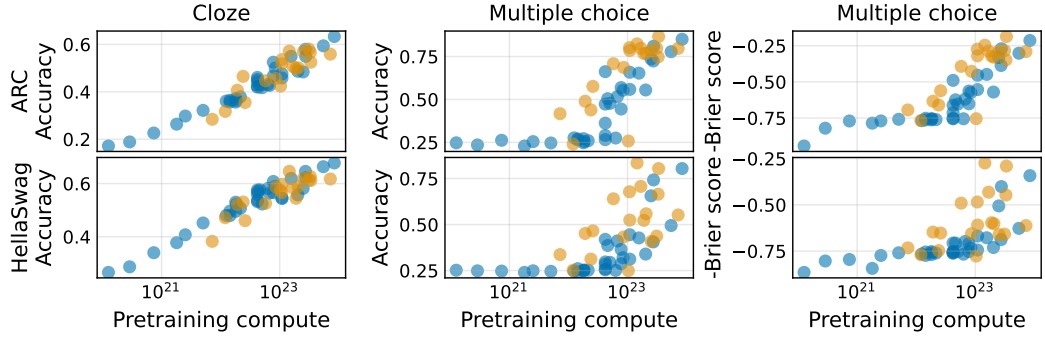

Figure 18: ARC and HellaSwag scores of models trained (●) before November 2023 and (●) after. *Middle*: reformulating the test task as multiple-choice leads to emergence around $10^{22}$ to $10^{23}$ FLOPs. *Right*: when using Brier score as the metric, we similarly observe sharp changes in performance around $10^{22}$ to $10^{23}$ FLOPs.

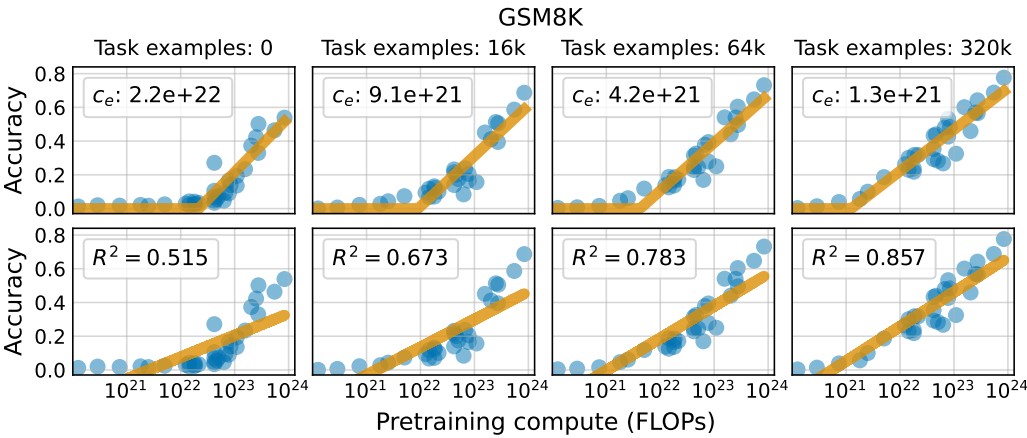

Figure 19: Scaling on GSM8K as models increasingly train on the test task. The point of emergence $c_e$ arises at lower scales (*top*). Training on the test task yields cleaner log-linear scaling fits (*bottom*).

# G  RESULTS FOR INSTRUCTION-TUNED AND CHAT MODELS

We evaluated 36 instruct and chat models, see Appendix B.1.1. Our findings for base models presented in the main text generalize remarkably well to instruction-tuned and chat models, see Figure 20. Newer instruct/chat models substantially outperform older instruct/chat models. However, after fine-tuning all models on the same amount of task-specific data, performance between newer and older instruct/chat models equalizes.

We find that the performance gap between newer and older instruct/chat models is smaller than the gap between newer and older base models, contrast the estimated effect sizes in Figure 1 with Figure 20). This is perhaps to be expected, as instruction-tuning datasets usually include some amount of benchmark data.

We posit that the gap between newer and older instruct/chat models is nonetheless large because early-to-mid 2023 instruction-tuned variants —e.g., Vicuna (Chiang et al., 2023), Alpaca (Taori et al., 2023), Llama 2 Chat (Touvron et al., 2023b)— did not emphasize benchmark performance but rather human preference (e.g., "win-rate") in a chat setting. For example, the Llama 2 technical report (Touvron et al., 2023b) includes no benchmark evaluations for their chat models. This perspective has dramatically shifted in the last year and a half, and post-training interventions now explicitly aim to improve benchmark performance (MetaAI, 2024; Gemma et al., 2024; Lambert et al., 2024).

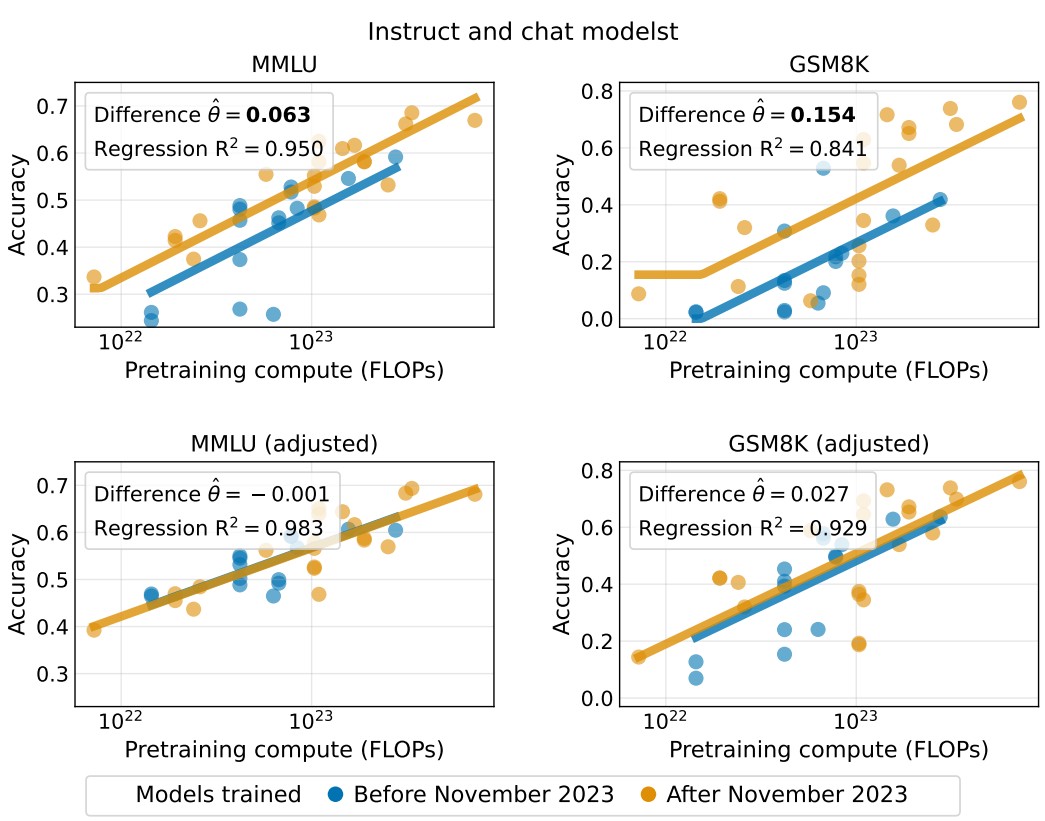

Figure 20: Reproducing Figure 1 for instruction-tuned and chat models.

