# OpenReview forum: "Training on the Test Task Confounds Evaluation and Emergence"
_ICLR.cc/2025/Conference — ICLR 2025 Oral_

### Official Review · Reviewer_HVsB · 2024-10-24

**Soundness:** 3
**Presentation:** 4
**Contribution:** 3
**Rating:** 8
**Confidence:** 4

**Summary:**

### Paper Summary
This paper asks if recent open-source models are in fact better than previous models: Does exposure to task data during pre-training explain the difference in model performance? (Yes). The authors find evidence that the difference on task performance can be explained by exposure to these tasks during the end of pre-training, not that more recent models are fundamentally better LLMs.

### Review Summary
There could be some world where this paper is simply stating the known fact that most models adjust the type of training data towards the end of pre-training to include task-relevant data, and that that inclusion improves performance. However, I think this paper does a good job of showing that this inclusion in particular seems to be making a key/big difference in models' improved performance, and that the inclusion during pre-training itself (while its being mixed with other data) doesn't seem to have an outsized impact on the ability to answer task-relevant questions.

**Strengths:**

The authors have a number of nice (creative/thoughtful) experiments showing how older models do equally well when given more task data, that changing the style of the question (not content) also closes the gap between models, and finally, that, when given more data, model families all seem to follow the same scaling law.

The figures are all great/clean/pretty/legible!

The writing is also all clear/careful!

**Weaknesses:**

* some could say that this is an obvious finding / direct outcome of how models recently have been trained. I don't hold this view. I find it interesting that there is not an additional/additive benefit found in recent models (from adding task related data in the end of pretraining), and interesting that the results are so stark. [i don't read your paper as saying that there is no gain/improvement in recent models, but rather, that performance doesn't extend beyond what fine-tuning directly upon the task in question would yield]
* figure 3 is unclear. the added information/gain of figure 3 vs figure 1 is unclear, esp. when looking at figure 3 alone (without referring to the text). to start, if I understanding things correctly, I think the colors of the points/lines should not be the same as the other figures as the underlying data is different from the other figures. the shared colors (to me) is misleading.

### nitpicks
* line 244, "These results provides..." --> "These results provide..."
* line 259, "contrast Figure 3 and Figure 1 Quantitatively..." --> "contrast Figure 3 and Figure 1. Quantitatively..."
* consider turning "What does MMLU test for?" into an active statement / takeaway.
* line 429 R^2 is formatted differently than line 430
* line 37, end of second paragraph. I was tripped up by "of some" and "of others". I now believe it is "of some models" / "of other models". I would guess the elisions were made to shorten/clean the paragraph? When I first read it I thought it was a typo that meant "to some people" / "to other people" (like different people will interpret this phenomena differently.)

**Questions:**

*  I didn't fully follow the final paragraph in section 4.2. Would your results not give an answer to your posed question? (i.e., doesn't task-training explain the improvements, and fall outside the scaling laws?)
* What eq do you fit in figure 8 bottom? Switching the equation used in the top row the one used in the bottom row didn't seem to be motivated in the text. Doesn't eq 1 also a capture a log-linear relationship? Spelling a bit more out here would help.
    * Giving a hint in the caption of figure 8 that the data are the same in both rows might help
    * Labelling the rows within the figure (not only the caption) might also help (like you do in figure 4).

---

> ### Author Response · Authors · 2024-11-22
>
> We thank you for your thoughtful review.
>
> > coloring in Figure 3
>
> We used similar color schemes for Figure 1 and Figure 3 to better convey the message that the difference in performance-per-computer between older and newer models strongly resembles that between older models and older models fine-tuned on task data. Indeed, at a glance Figure 1 and Figure 3 look very similar – this is because the performance-per-compute of newer models resembles that of older models fine-tuned on task data, both pre and post adjustment. The blue points on both plots are the same: the performance of older models. The orange points differ: In Figure 1 it is the performance of newer models, whereas in Figure 3 it is the performance of older models that were fine-tuned on the task data for one epoch.
>
> > Regression fit in Figure 8 bottom
>
> The regression fit in figure 8 bottom is a log-linear fit rather than the piece-wise log-linear “hinge fit” of figure 8 top, which we describe in Equation 1. The goodness-of-fit of the log-linear fit conveys that fine-tuning on task data leads to cleaner log-linear scaling behavior: almost all of the variation in benchmark accuracy is explained by log-linear scaling of pre-training compute alone. This does not directly follow from the top plot, which instead conveys that the point of emergence occurs at smaller compute scales as one increasingly trains on task data. While both phenomena are closely related, we believe that separately presenting the two regression fits allows us to more clearly convey both points.
>
> > Final paragraph in Section 4.2.
>
> Chinchilla scaling laws predict that some pre-training compute allocations (that is, the number of model parameters relative to the number of pre-training tokens) are more favorable than others. When trained on the same data distribution, Llama 2 70B should attain lower loss than Llama 3 8B according to the Chinchilla scaling laws. It is therefore somewhat surprising that, despite having similar pre-training compute and controlling for training on the test task, Llama 3 8B closely matches the performance of Llama 2 70B.

---

### Official Review · Reviewer_b3hy · 2024-10-26

**Soundness:** 3
**Presentation:** 4
**Contribution:** 3
**Rating:** 8
**Confidence:** 4

**Summary:**

The paper argues that newer language models only appear to be better on metrics like MMLU than earlier, iso-compute language models because of the presence of data in distribution to MMLU in the pretraining set. The paper demonstrates this by showing that when these models are finetuned on MMLU/GSM8K before evaluation, the performance differences between different model families go away. The paper also shows that emergent behaviors are less common under this new evaluation protocol.

**Strengths:**

- The paper is very clearly presented and argues persuasively. The figures are easily understood at a glance.
- The conclusion argued for is surprising and novel, and has broad implications for evaluation methodology of LLMs and emergent behaviors in LLMs.

**Weaknesses:**

- The p-values for signficance in figure 1 overestimate significance because of correlation due to models being from only a few model families.
- The paper never actually directly checks the hypothesis of more in-distribution data being the cause of higher MMLU performance out of the box, despite arguing for it via implication from the finetuning result.

**Questions:**

I'd be interested to see (on open source models with available datasets, or by sampling unconditionally from base models) whether there is more MMLU-like data in model families which appear to do better on MMLU when using the naive evaluation method.

---

> ### Author Response · Authors · 2024-11-22
>
> We thank you for your thoughtful review.
> > The p-values for signficance in figure 1 overestimate significance because of correlation due to models being from only a few model families.
>
> We have re-run the significance tests using OLS with clustered standard errors, where we consider each model family a different group. While the p-values of course change, we get identical significance results. We will use this significance test moving forward. Thank you for raising this point.
>
> > The paper never actually directly checks the hypothesis of more in-distribution data being the cause of higher MMLU performance out of the box. I'd be interested to see (on open source models with available datasets, or by sampling unconditionally from base models) whether there is more MMLU-like data in model families which appear to do better on MMLU when using the naive evaluation method.
>
> Most model families released after November 2023 are explicit about the use of benchmark data at pre-training time, see the general response. Open models with publicly available datasets are even more explicit on the use of instruction-data: StableLM 2 and Olmo 1.7 both include the FLAN instruction dataset in the pre-training data, which contains part of the MMLU auxiliary training data. Thus, we know with certainty that these two model families saw at least part of the MMLU auxiliary training set.
>
> Directly determining whether such instruction-data is the cause of their higher MMLU performance would require re-training the models, with all being equal except for excluding such instruction datasets. This is computationally infeasible. Nonetheless, Olmo presents an interesting case study. The main differentiating factor between Olmo 1.0 and Olmo 1.7 (both trained early 2024) is their pre-training data, and in particular including “content specifically sourced to improve model performance on tasks requiring specialized knowledge (e.g. arXiv, Stack Exchange) and complex reasoning (e.g. OpenWebMath, **Flan**)” [1]. While Olmo 1.0 7B performs roughly random-chance on MMLU, Olmo 1.7 7B attains a 24 point improvement on MMLU.
>
> [1] AllenAI, OLMo 1.7–7B: A 24 point improvement on MMLU, https://allenai.org/blog/olmo-1-7-7b-a-24-point-improvement-on-mmlu-92b43f7d269d. 2024.

---

### Official Review · Reviewer_xmL1 · 2024-11-03

**Soundness:** 3
**Presentation:** 3
**Contribution:** 3
**Rating:** 8
**Confidence:** 4

**Summary:**

The paper explores an interesting phenomenon: training on the test task, which differs from training on the test data. Training on test tasks implies that task-relevant data is included in the pretraining stage of a language model. The paper highlights how newer models benefit from training on the test task, but when fine-tuning older models on task-relevant data, this performance advantage disappears.

**Strengths:**

- This paper explores an ever-increasingly important area of LLMs: the "science of evaluations."
- It explores the implications of task-specific training for language models, providing evidence that training with task-specific data is beneficial. They find that many models from November 2023 gain advantages from this type of training. Additionally, they show that task-specific training is more likely responsible for recent boosts in model performance when FLOPs are held constant.
- The paper's implications (as stated in the discussion) suggest that training on task-specific data may be beneficial across models.
- The experiments and the paper are logical and well thought through.

**Weaknesses:**

- How was November 2023 chosen as the cutoff? I wonder what would have happened if you had chosen November 2022. How would the results be affected?
- It is unclear how much hyperparameter tuning might affect results when training on task-specific data. How much could hparam tuning affect the results? How was the initial sweep in Appendix A.2 chosen? Although a full sweep might be expensive, additional clarity on how these were chosen will alleviate my concern.
- Lack of consideration of instruction-tuning models. How do you think instruction tuning models might impact this study as often the newer datasets are comprised of examples to explicitly increase benchmark performance [1]?
- Only a couple of main evaluation datasets are explored in the paper.

[1] GenQA: Generating Millions of Instructions from a Handful of Prompts

**Questions:**

- Do you the authors think LLMs should be part of the title as only LLMs are explored in the paper?

---

> ### Author Response · Authors · 2024-11-22
>
> Thank you for your thoughtful review.
>
> > Lack of consideration of instruction-tuning models.
>
> To address this limitation, we have conducted substantial additional experiments, please see the general response.
>
> > How was November 2023 chosen as the cutoff? I wonder what would have happened if you had chosen November 2022. How would the results be affected?
>
> We chose November 2023 as the cut-off because models released in late 2023 start to mention the use of instruction data at pre-training time, please see the general response. We additionally show the results when choosing November 2022 as the cut-off point in Figure 19. Here, the blue points correspond to the Pythia model family. Models newer than Pythia outperform Pythia. However, after giving all models the same amount of task specific data, we find no evidence for a significant difference in performance between the Pythia model family and all other models. This is consistent with our extended discussion in Section 4.1 “Comparing model families”. Note that the pre-training dataset of Pythia is the Pile, a collection of curated datasets that are unlikely to contain much test task data.
>
> We also find it illuminating to contrast the performance of only model families released after November 2023, that is, around the time that models start to report actively using task data at pre-training time. Since we compare models released between November 2023 and May 2024, we choose February 2024 as the cut-off point. We plot the results in Figure 20. We find no evidence of a significant difference in performance between models.
>
> > How much could hparam tuning affect the results? How was the initial sweep in Appendix A.2 chosen?
>
> We chose the initial hyperparameters according to our prior experience fine-tuning language models. We performed minimal tuning due to its expensive nature. We found the learning rate to be, by a large margin, the single most impactful hyperparameter. To alleviate concerns regarding the potential impact of poor hyperparameter tuning on the results, we have performed a sweep on MMLU with the following learning rates: [6e-5, 2e-5, 6e-6, 2e-6, 6e-7].  We are unable to perform more extensive sweeps due to their large computational cost. Note that we originally used 2e-5 for most models, and 2e-6 for models for which 2e-5 lead to instability. We fine-tuned a total of 50x4=200 additional models for such a sweep. No models benefit from smaller learning rates than 2e-6, and only one model benefits from larger learning rate than 2e-5, Pythia 70M (the model with smallest pre-training compute). Thus, the optimal learning rate is inside the boundary of the sweep, except for Pythia 70M.
>
> Since the concern is that older models match the performance of newer models because the hyperparameters used may be systematically more favorable to older models, we consider the following: we recreate our main experiment; with older models using the originally chosen learning rate, but choosing for the newer models the learning rate that leads to highest MMLU performance after fine-tuning. That is, we give newer models a systematic advantage in terms of hyperparameter search budget. We plot the results in Figure 21. The measured effect size of model recency on benchmark performance changes from $\hat{\theta}=-0.005$ to $\hat{\theta}=0.005$. Therefore, after giving new models a systemic advantage in terms of hyperparameter search budget, the effect size remains small. It is also not statistically significant.
>
> >Only a couple of main evaluation datasets are explored in the paper.
>
> We consider the following 10 benchmarks: MMLU, GSM8K, ARC-Challenge, and HellaSwag (from the HuggingFace Leaderboard v1, Section 4.2) as well as MMLU Pro, GPQA, MuSR, BBH, and MATH Lvl 5 (from the HuggingFace Leaderboard v2, Appendix C). We pay particular attention to MMLU and GSM8K for two reasons: 1) because they are the two most influential benchmarks of the 2023-2024 period that we study, and 2) because, remarkably, we find strong empirical evidence of the effect of training on the test task on such highly influential benchmarks. However, we show that the phenomenon of training on the test task in not specific to MMLU and GSM8K, see our extended discussion in Appendix C regarding the benchmarks of the HuggingFace Leaderboard v2.

---

> > ### Comment · Reviewer_xmL1 · 2024-11-26
> >
> > The authors have addressed my concerns, and I will be raising my score to an eight.

---

### Official Review · Reviewer_Zcyd · 2024-11-04

**Soundness:** 3
**Presentation:** 4
**Contribution:** 4
**Rating:** 8
**Confidence:** 3

**Summary:**

The paper defines *training on the test task* as a potential problem for evaluating LLMs, which could be common practice and strictly speaking not data contamination. They show that doing so may allow a model to achieve better results on a benchmark while not having better abilities overall. They suggest that further fine-tuning all models can help with better comparison across models, and show that doing so restores clearer scaling trends.

**Strengths:**

* They test their method on on a large set of base models with different sizes. This also allows them to compare scaling on benchmarks.
* The proposed method is promising for better evaluations/comparisons and light on compute, requiring only fine-tuning.

**Weaknesses:**

* They only do so for pre-trained base models. In practical settings, we often look at the benchmarks of fine-tuned chat models, or models further fine-tuned off base-models (e.g. LLama derivatives). It is not clear whether their findings would apply to already-finetuned models.
* They classify 'old' models as before November 2023, which seems somewhat arbitrary. I understand that choosing Nov 2023 reveals some sort of improvement given the same compute, but this could be due to reasons other than training on the test task. More justification is needed on why the improvement gap is a good heuristic for whether a model was trained on the test task. Alternative analysis on whether much more models after Nov 2023 include instruction data in the pre-training set could similarly justify why Nov 2023 is a good cut-off point.

**Questions:**

* Ultimately, benchmarks are used to understand how good a model is at a general skill, like mathematical reasoning for MMLU and GSM8K. This method proposes to 'fight fire with fire' by fine tuning all models on the test task. Would your method merely reliably measure how well models performs on a specific benchmark, rather than how well models are at the ability it's meant to measure?

---

> ### Author Response · Authors · 2024-11-22
>
> Thank you for the thoughtful review.
>
> > It is not clear whether their findings would apply to already fine-tuned models.
>
> To address this limitation, we have conducted substantial additional experiments, please see the general response.
>
> > Alternative analysis on whether much more models after Nov 2023 include instruction data in the pre-training set could similarly justify why Nov 2023 is a good cut-off point.
>
> Please see the general response. Indeed, most models released after November 2023 use instruction data at pre-training time. We verify that shifting the cut-off by a few months (e.g., September 2023 or January 2024) leads to very similar results, see Figure 10 and Figure 11 in the Appendix.
>
> > Ultimately, benchmarks are used to understand how good a model is at a general skill, like mathematical reasoning for MMLU and GSM8K. [...] Would your method merely reliably measure how well models perform on a specific benchmark, rather than how well models are at the ability it's meant to measure?
>
> The main purpose of machine learning benchmarks is the relative comparison of models under standardized conditions. They enable the competitive empirical testing that ultimately drives scientific progress in our field [1, 2]. Our proposed benchmarking methodology ensures fair relative model comparisons in light of the disproportionate large effect that task data has on benchmark evaluations. Whether and the extent to which benchmark evaluations may be indicative of broader “abilities” is a much debated topic (e.g., [3, 4, 5]) that is tangential to our work.
>
> [1] Liberman, Mark. "Obituary: Fred Jelinek." Computational Linguistics 36.4 (2010).
>
> [2] Hardt, Moritz. “The Emergening Science of Benchmarks”. ICLR Invited Talk (2024).
>
> [3] Bowman, Samuel, and George Dahl. "What Will it Take to Fix Benchmarking in Natural Language Understanding?." NAACL HLT (2021).
>
> [4] Raji, Inioluwa Deborah, et al. "AI and the everything in the whole wide world benchmark." NeurIPS 2021 Benchmarks and Datasets track (2021).
>
> [5] Liang, Percy, et al. "Holistic evaluation of language models." TMLR. (2023).

---

> > ### Comment · Reviewer_Zcyd · 2024-11-26
> >
> > I thank the authors for their response.
> >
> > Overall, I find the idea to be simple and enticing. Given that the authors have further validated their methodology especially by doing further experiments on my two main concerns, I have raised my score to an 8.

---

### Author Response · Authors · 2024-11-22

We thank the reviewers. We will address in the general response the concerns raised by both reviewer Zcyd and reviewer xmL1.
### Instruction-tuned models
We have substantially expanded our experiments by considering an additional 36 “chat” and “instruct” language models, see Appendix E.1 of the revised manuscript. Our original findings generalize remarkably well, see Figure 18 in Appendix E.1. Newer instruct/chat models substantially outperform older instruct/chat models. After fine-tuning all models on the same amount of task-specific data, performance between newer and older instruct/chat models equalizes.

We find that the performance gap between instruct/chat models is smaller than the gap between base models (that is, $\hat{\theta}$ in Figure 1 vs Figure 18). This is unsurprising, as instruction-tuning datasets usually include some amount of benchmark data (e.g., [1]). We posit that the gap between newer and older instruc/chat models is nonetheless large because early-to-mid 2023 instruction-tuned variants (e.g., Vicuna, Alpaca, Koala, Llama 2 Chat) did not emphasize benchmark performance but rather human preference (e.g., “win-rate”) in a chat setting. For example, the Llama 2 technical report [2] includes no benchmark evaluations of their fine-tuned and RLHF’d models. This perspective has dramatically shifted in the last year and a half, and post-training interventions now explicitly aim to improve benchmark performance (e.g., [3, 4]).

### Choosing November 2023 as the cutoff

We chose November 2023 as a cut-off because the technical reports of models released starting late 2023 begin to mention the use of instruction data for training. We carefully examined the technical report of all model families evaluated. Among these models, none released before November 2023 mention the use of instruction data for pre-training. In contrast, most model families released after November 2023 mention doing so. Qwen 1.5, Olmo 1.7, StableLM 2, and MAP Neo explicitly include instruction data in the pre-training mixture. Llama 3 and Gemma state that their pre-training data was determined through ablations on benchmark evaluations, a mechanism through which models may implicitly train on the test task [5].

Our analysis is robust to shifting the temporal split by a few months. We include such a robustness analysis in Appendix B.2., specifically Figure 10 which uses Sept 2023 as the cut-off month and Figure 11 which uses Jan 2024 as the cut-off month. Our findings indicate that practitioners started adopting design choices around late 2023 that amount to models training on the test task more, which is consistent with models’ technical reports starting to mention the use of benchmark or instruction data for training. Choosing specifically the month of November as the cut-off is therefore not critical for our analysis.

[1] Mukherjee, Subhabrata, et al. "Orca: Progressive learning from complex explanation traces of gpt-4." arXiv preprint arXiv:2306.02707 (2023).

[2] Touvron, Hugo, et al. "Llama 2: Open foundation and fine-tuned chat models." arXiv preprint arXiv:2307.09288 (2023).

[3] Team, Gemma, et al. "Gemma: Open models based on gemini research and technology." arXiv preprint arXiv:2403.08295 (2024).

[4] Dubey, Abhimanyu, et al. "The llama 3 herd of models." arXiv preprint arXiv:2407.21783 (2024).

[5] Hewitt, John, et al. "Instruction following without instruction tuning." arXiv preprint arXiv:2409.14254 (2024).Z

---

### Meta-Review · Area_Chair_MUHn · 2024-12-19

**Metareview:**

The authors describe a confounder in evaluations, where the ability of a model to follow the task format is mixed with its underlying ability to do a task - for example, in a task like MMLU, the knowledge of the model is confounded with its ability to do few-shot multiple-choice tasks. The authors propose fine-tuning on a small, common set of task-related data to put all the methods on equal footing, which makes evaluation more stable gives consistent measurements even at small scales (handling emergence).

This is a timely, important problem and the fine-tuning approach is novel (though I think concurrent with Snell et al at COLM) and the reviewer comments are addressed clearly.

**Additional Comments On Reviewer Discussion:**

The reviewers raised some nice technical points - like the computation of p-values, as well as some basic clarification questions about cutoff points. All of these have been handled clearly and directly.

---

### Decision · Program_Chairs · 2025-01-22

Accept (Oral)